# SolidGen: An Autoregressive Model for Direct B-rep Synthesis

**Pradeep Kumar Jayaraman**                                    *pradeep.kumar.jayaraman@autodesk.com*
*Autodesk Research*

**Joseph G. Lambourne**                                          *joseph.lambourne@autodesk.com*
*Autodesk Research*

**Nishkrit Desai**                                              *nishkrit.desai@mail.utoronto.ca*
*University of Toronto, Vector Institute*

**Karl D.D. Willis**                                                    *karl.willis@autodesk.com*
*Autodesk Research*

**Aditya Sanghi**                                                      *aditya.sanghi@autodesk.com*
*Autodesk Research*

**Nigel J.W. Morris**                                                  *nigel.morris@autodesk.com*
*Autodesk Research*

**Reviewed on OpenReview:** *https://openreview.net/forum?id=ZR2CDgADRo*

## Abstract

The Boundary representation (B-rep) format is the de-facto shape representation in computer-aided design (CAD) to model solid and sheet objects. Recent approaches to generating CAD models have focused on learning sketch-and-extrude modeling sequences that are executed by a solid modeling kernel in postprocess to recover a B-rep. In this paper we present a new approach that enables learning from and synthesizing B-reps without the need for supervision through CAD modeling sequence data. Our method SolidGen, is an autoregressive neural network that models the B-rep directly by predicting the vertices, edges, and faces using Transformer-based and pointer neural networks. Key to achieving this is our Indexed Boundary Representation that references B-rep vertices, edges and faces in a well-defined hierarchy to capture the geometric and topological relations suitable for use with machine learning. SolidGen can be easily conditioned on contexts e.g., class labels, images, and voxels thanks to its probabilistic modeling of the B-rep distribution. We demonstrate qualitatively, quantitatively, and through perceptual evaluation by human subjects that SolidGen can produce high quality, realistic CAD models.

## 1 Introduction

Almost every manufactured object in the world starts life as a computer-aided design (CAD) model. The Boundary representation format (B-rep) is the de-facto standard used in CAD to represent solid and sheet objects as a collection of trimmed parametric surfaces connected with well-structured topology (Weiler, 1986). The ability to generate B-reps automatically, driven by some context, is an enabling technology for design exploration and is critical to solving a range of problems in CAD such as reverse engineering from noisy 3D scan data, inpainting holes in solid models plausibly, and design synthesis from images or drawings.

Recent approaches to the generation of CAD models have focused on generating sequences of CAD modeling operations (Willis et al., 2021a; Xu et al., 2021; Wu et al., 2021; Xu et al., 2022; Lambourne et al., 2022),

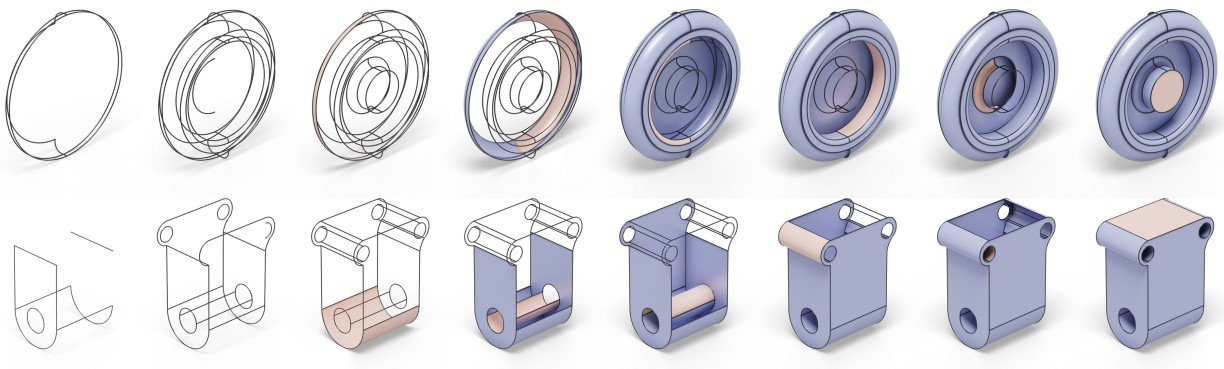

Figure 1: SolidGen generates B-reps incrementally by building vertices, edges, and faces one at a time. Here we show snapshots from the edge (columns 1–2) and face generation (columns 3–8) for two data samples.

rather than the underlying 3D geometry and topology in the B-rep format. These methods produce a sequence of sketch and extrude modeling operations using a neural network and the B-rep is recovered in postprocess with a solid modeling kernel that executes the operations. Although this approach generates an editable CAD model, it is currently limited to the sketch and extrude modeling operations and cannot be easily extended to support other operations or build sheet bodies. In particular, fillets and chamfers, which are widely used in CAD for structural performance and ease of manufacture, are challenging as they operate on B-rep edges which are not available until the predicted solid has been built. Furthermore, sketch-and-extrude workflows operate with 2D planar curves, hence extensions to freeform modeling are not trivial.

There are several advantages to pursuing an approach that directly synthesizes the B-rep. Firstly, significantly more CAD data exists without a stored sequence of CAD modeling operations. Files created via direct modeling or exported to common B-rep file formats do not retain a history of CAD modeling operations. In the public domain, CAD model datasets with CAD modeling operations (Willis et al., 2021a; Wu et al., 2021) are smaller (∼190K models) than those without (Koch et al., 2019) (1M models). Secondly, support for freeform curves and surfaces like Bézier and non-uniform rational B-splines, or advanced topological structures like T-splines or Catmull-Clark subdivision meshes can be provided with the ability to generate B-reps directly, as they share geometric and topological similarities. Finally, there are several problems in CAD that can only be solved with a direct B-rep synthesis approach. Examples include hole-filling for repairing solids which have poorly trimmed or missing faces due to data exchange (Butlin & Stops, 1996; Assadi, 2003), patching up geometry in regions of a model where error-prone operations, such as offsetting, fail (Bodily, 2014) and the creation of parting surfaces and shut-outs in molding workflows (Bhargava et al., 1991; Ser, 1995). Although heuristic algorithms exist for such applications, a learning-based method has the potential to incorporate external context and respect aesthetic aspects of the CAD model.

We make the following contributions in this paper:

- We propose SolidGen, a novel generative model based on Transformers and two-level pointer networks for the direct synthesis of B-reps (Figure 1) without supervision from a sequence of CAD modeling operations. To the best of our knowledge, SolidGen is the first generative model that can synthesize B-reps directly.

- We propose a new representation, the indexed boundary representation, that can represent B-reps as numeric arrays suitable for use with machine learning, while still allowing the geometry and topology of B-reps to be completely recovered.

- We show the quantitative and qualitative performance of SolidGen for unconditional generation and perform a perceptual study showing SolidGen produces more realistic results than the state-of-the-art.

- We demonstrate how controllable generation of B-reps can be achieved by conditioning with class labels, images, and voxels when available.

## 2 Related Work

Since the proliferation of the B-rep format (Lee & Lee, 2001; Slyadnev et al., 2020; Weiler, 1986) in the 1980s, several research areas have been explored.

**Learning from B-reps.** The ability to represent B-rep data as a graph (Ansaldi et al., 1985) has inspired recent approaches using graph neural networks for classification and segmentation problems (Cao et al., 2020; Jayaraman et al., 2021). B-rep topology can also be used for custom convolutions (Lambourne et al., 2021) or hierarchical graph structures (Jones et al., 2021). Rather than an *encoder*, our work focuses on a *decoder* to synthesize B-rep data directly.

**Constructive Solid Geometry (CSG).** In CSG, 3D shapes are formed by combining primitives (e.g. cuboids, spheres) with Boolean operations (e.g. union, subtraction) into a CSG tree. By parsing the tree with an appropriate geometry kernel, a B-rep can be obtained. CSG approaches have been used to reconstruct 'shape programs' in combination with techniques from the program synthesis literature, both with neural guidance (Sharma et al., 2018; Ellis et al., 2019; Tian et al., 2019; Kania et al., 2020) and without (Du et al., 2018; Nandi et al., 2017; 2018). By contrast, our work applies to multiple tasks beyond reconstruction.

**Sequential CAD Generation.** Another line of research leverages the supervision available from modeling operations stored in parametric CAD files to directly predict sequences of CAD modeling operations. The result is editable CAD files in the form of 2D engineering sketches (Willis et al., 2021b; Para et al., 2021; Ganin et al., 2021; Seff et al., 2021) or 3D CAD models generated from a B-rep (Willis et al., 2021a; Xu et al., 2021) or point cloud (Uy et al., 2021) target, interactive system (Li et al., 2020), or generative model (Wu et al., 2021; Xu et al., 2022). Although sequential CAD generation approaches have made significant progress, an outstanding challenge is how to extend these techniques to other modeling operations beyond sketch and extrude. Common modeling operations, such as fillet and chamfer, are challenging as they operate on B-rep edges which are not part of the representation available to the network, but rather generated as a post-process. An alternate approach might be to use reinforcement learning with a CAD kernel integrated into the environment, similar to Lin et al. (2020), at the expense of extended training times due to the sparse reward space and slow CAD kernel execution times. Instead, we focus our efforts on direct synthesis of the B-rep data structure which cannot be achieved by predicting a sequence of CAD modeling operations.

**Direct B-rep Generation.** Direct B-rep generation involves the synthesis of the underlying parametric curves/surfaces and the topology that connects them to form a solid model. Several learning-based approaches have made progress with the generation of parametric curves (Wang et al., 2020) and surfaces (Sharma et al., 2020). Yet to be addressed is a generative model for the creation of topology that joins parametric curves and surfaces together to form solid models, rather than relying on pre-defined topological templates (Smirnov et al., 2021). PolyGen (Nash et al., 2020) is a promising approach to joint geometry and topology synthesis where sequences of $n$-gon mesh vertices and faces are predicted using Transformers (Vaswani et al., 2017). A key insight of this work is the use of pointer networks (Vinyals et al., 2015) to reference previously predicted primitives and form the underlying shape topology. Our work draws inspiration from the use of pointer networks to develop a novel representation and method for the challenging task of direct B-rep generation. Wang et al. (2022) concurrently applied a pointer network to identify planar and cylindrical faces given the 2D projection of edges. Our method is more general and synthesizes the entire B-rep while supporting all prismatic surfaces in a generative modeling framework. A similar representation to ours was proposed in a concurrent work (Guo et al., 2022), where the objective was to reverse engineer B-reps from point clouds. Using a neural network to predict the rough B-rep geometry and topology, a combinatorial optimization was applied to refine the solution. By contrast, our method is an autoregressive generative model with support for more input modalities such as image and class in addition to point clouds, and does not require an expensive optimization step to build plausible B-reps.

## 3 Representation

**Boundary Representation.** B-reps are a kind of generalized mesh structure in which many of the restrictions of triangle meshes have been removed (Figure 2). In place of planar triangular facets, B-reps allow *faces* built from planes, cylinders, cones, spheres, toroidal surfaces or B-splines surfaces. The *edges*

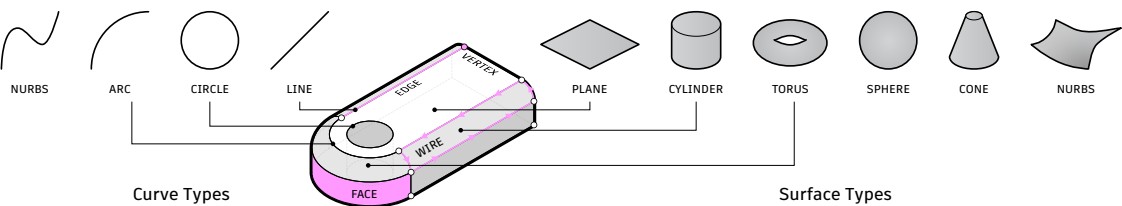

Figure 2: The Boundary Representation data structure consists of vertices, edges formed by curve primitives (left) and faces formed by surface primitives (right). Loops of oriented edges—wires, are used to delimit the surfaces into visible and hidden regions.

at the boundary of faces can be lines, arcs, circles or B-spline curves. Unlike meshes, B-rep faces can have internal holes, hence closed loops of edges are organized into oriented *wires* to delimit the surfaces. The curves defining the edges can be viewed as functions from an interval in the 1D u-parameter domain into 3D, while the surfaces can be viewed as functions from a 2D box in the uv-parameter domain to 3D. The wires defining a face's boundaries can be projected onto its surface's 2D parameter domain, forming closed loops. Outer loops in uv-space are oriented anti-clockwise, while inner loops are in clockwise direction. This allows uv-points to be classified as visible or hidden and facilitates meshing (Chew, 1987; Rockwood et al., 1989) [1]. The B-rep data structure stores many references among entities (typically using pointers) allowing efficient constant-time querying of adjacency information (Lee & Lee, 2001).

**Indexed Boundary Representation.** We now derive an indexed representation of the B-rep (Figure 3, right) that can be represented as numeric arrays suitable for use with machine learning. Importantly, this representation can be inverted back into a B-rep exactly. We currently restrict the representation to elementary curves (lines and arcs) and surfaces (planes, cylinders, cones, spheres and tori). We do not support B-splines and conic sections that are uncommon in mechanical CAD (Willis et al., 2021a; Koch et al., 2019). Without loss of generality, we assume that closed faces (cylinders, cones, etc.) in the B-rep are all split on the seams to simplify downstream processing, for e.g., a closed cylinder with a seam is split into two four sided half-cylinders. The indexed B-rep is analogous to the indexed list data structure used to represent polygonal meshes, e.g. the Wavefront OBJ file format (Library of Congress, 2020). The indexed list data structure can be represented numerically using two lists of numbers, and has been successfully used for generative modeling of meshes (Nash et al., 2020) and engineering sketches (Willis et al., 2021b). Just as a polygonal mesh can be represented using a topological data structure like winged edge (Baumgart, 1972), or a simple data structure like the indexed list, we show that B-reps can be converted into an indexed format. However, while a polygonal mesh is restricted to straight line edges and planar faces, B-reps are more general with support for curved edges and faces, making our problem more challenging. We address this challenge by explicitly representing only the bare minimum geometric and topological information in our indexed format, and designing a rule-based algorithm leveraging solid modeling kernels to infer and build the rest of the information. The indexed B-rep $\mathcal{B}$ is comprised of three lists $\{\mathcal{V}, \mathcal{E}, \mathcal{F}\}$: *Vertices* $\mathcal{V}$ contain a list of 3D point coordinates corresponding to each B-rep vertex and the arc midpoints. *Edges* $\mathcal{E}$ are represented as hyperedges (Willis et al., 2021b) where each hyperedge is a list of indices into $\mathcal{V}$. A hyperedge connects two or more vertices to define the edge geometry. The curve type is defined by the cardinality of the hyperedge: line (2), arc (3). *Faces* $\mathcal{F}$ are defined as the set of edges bounding a surface. Each face contains a list of indices into $\mathcal{E}$. By indexing from *Faces*, to *Edges*, to *Vertices*, our representation is well suited for use with pointer networks (Vinyals et al., 2015). Parametric surfaces that define the geometry of the face, and wires that trim the surfaces are left out of our representation and recovered in postprocess, as described next.

**Recovering a Boundary Representation.** We leverage the boundary curve geometry that is defined by the edges $\mathcal{E}$, together with the mapping of the edges that bound a face $\mathcal{F}$ to recover the wire and surface geometry using well-defined rules. In particular, the surface type can be first fixed from the curve types

---

[1] For periodic surfaces (cylinders, cones, spheres and tori), which have parameter domains "topologically glued" from 0 to $2\pi$ in one or both dimensions, the surface must first be subdivided into patches homeomorphic to the disk.

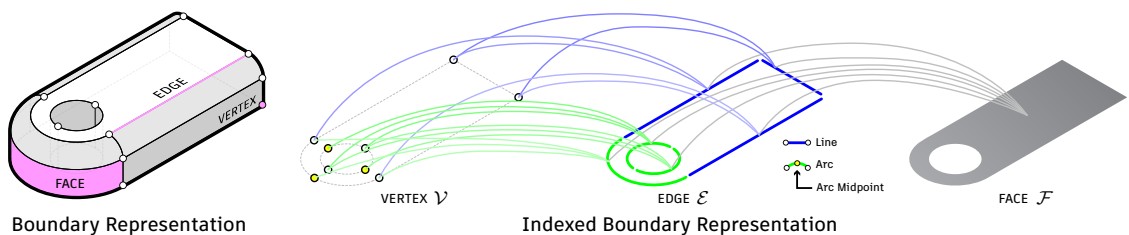

Figure 3: The Boundary Representation (left) comprises several topological entities. Our Indexed Boundary Representation (right) references vertices, edges, and faces in a well-defined hierarchy suitable for use with machine learning.

while the surface parameters can be inferred from the curve and vertex geometry. The surface type and parameters can be derived from the curves by a simple set of rules. For instance, faces whose vertices are coplanar define a trimmed planar surface, faces whose curves are a non-planar collection of lines and arcs will be cylinders or cones and faces whose curves are all arcs will be spheres or tori. We explain the procedure to apply this rule-based algorithm and build the B-rep in greater detail in Section A.2. Applying this procedure to every face in the indexed B-rep yields the surface type and parameters. Then, the data is read into the solid modeling kernel using standard data exchange and shape fixing functions designed for reading neutral CAD file formats like STEP and IGES.

## 4 SolidGen Architecture

SolidGen is a generative model that directly synthesizes CAD data in the B-rep format. SolidGen leverages Transformers (Vaswani et al., 2017) and pointer networks (Vinyals et al., 2015) to enable the generation of vertices, curved edges, and non-planar faces, such that edges can refer to vertices and faces can refer to edges, allowing us to build the complete B-rep topology. Our goal is to estimate a distribution over indexed B-reps $\mathcal{B}$ that can be sampled from to generate new indexed B-reps, and converted to actual B-reps as described in the previous section. This probability distribution $p(\mathcal{B})$ is defined as the joint distribution of the vertices $\mathcal{V}$, edges $\mathcal{E}$, and faces $\mathcal{F}$

$$p(\mathcal{B}) = p(\mathcal{V}, \mathcal{E}, \mathcal{F}). \tag{1}$$

To make the learning problem tractable, we factorize this joint distribution into a product of conditionals

$$p(\mathcal{B}) = p(\mathcal{F}|\mathcal{E}, \mathcal{V})p(\mathcal{E}|\mathcal{V})p(\mathcal{V}), \tag{2}$$

and learn these distributions with separate neural networks. Once these conditional distributions are learned, B-reps can be generated by sampling vertices first, followed by edges conditioned on vertices, and faces conditioned on the edges and vertices. It is also possible to condition the generation using a context $z$ in which case the joint distribution becomes

$$p(\mathcal{B}|z) = p(\mathcal{F}|\mathcal{E}, \mathcal{V}, z)p(\mathcal{E}|\mathcal{V}, z)p(\mathcal{V}|z), \tag{3}$$

where $z$ can be derived from other representations like class labels, images, voxels, etc. Since there can be an arbitrary number of vertices, edges, and faces in each B-rep, and strong symmetries are present in CAD models, we use autoregressive neural networks to model the probability distributions above. Figure 4 shows an overview of the SolidGen architecture at inference time, with each model described in further detail below.

### 4.1 Vertex Model

The vertex model (Figure 4, left) learns a distribution $p(\mathcal{V})$ over the vertex positions. Vertices here include both B-rep vertices and the additional points inserted on B-rep edges to encode the curve primitive information as explained in Figure 3. The vertices are first sorted lexicographically, i.e., by z coordinates first, y coordinates next, and x coordinates finally, then their coordinates are flattened into a 1D list, with a stopping token `<EOS>` marking the end. After flattening, $\mathcal{V}^{\text{seq}} = \{v_0, v_1, \ldots, \texttt{<EOS>}\}$ is of length $|\mathcal{V}^{\text{seq}}| = (|\mathcal{V}| \times 3) + 1$,

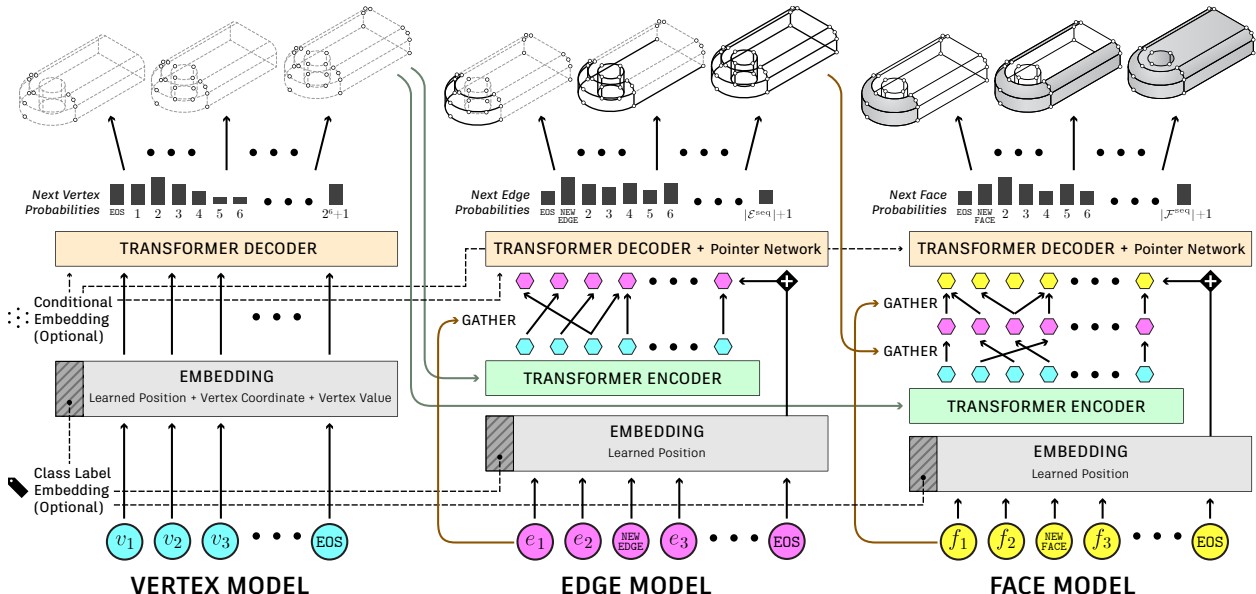

Figure 4: The SolidGen architecture consists of a vertex (left), edge (center), and face (right) model that predict the vertices, edge connections between vertices, and face connections between edges to form an indexed B-rep (top).

with every triplet corresponding to coordinate values of a single vertex. Rather than using real-valued coordinates for the vertices, we quantize the vertex sequence $\mathcal{V}^{\text{seq}}$ uniformly into 6-bits as is common in previous work (Nash et al., 2020; Seff et al., 2021; Wu et al., 2021). The vertex model learns the following distribution

$$p(\mathcal{V}^{\text{seq}}; \theta_{\mathcal{V}}) = \prod_{t=0}^{|\mathcal{V}^{\text{seq}}|-1} p(v_t \mid v_0, v_1, \ldots, v_{t-1}; \theta_{\mathcal{V}}), \qquad (4)$$

where $\theta_{\mathcal{V}}$ are learnable parameters of the neural network that is a Transformer decoder (Vaswani et al., 2017), and the conditional distributions above can all be treated as categorical distributions.

Given the input vertices $\mathcal{V}^{\text{seq}}_{<t}$ at step $t$, the goal is to model the next output vertex token $v_t$. The input to the Transformer decoder $\mathbf{h}_{\mathcal{V}}$ is derived from each of the tokens in $\mathcal{V}^{\text{seq}}_{<t}$ by learning three kinds of embeddings— *coordinate embeddings* indicating whether a token is an x, y or z-coordinate, *positional embeddings* indicating the vertex index which the token belongs to, and a *value embedding* encoding the x, y or z-coordinate:

$$\mathbf{h}_{\mathcal{V}^{\text{seq}}_{<t}} = \{\mathbf{W}_{\text{coo}}(\mathbb{1}_{i \bmod 3}) + \mathbf{W}_{\text{pos}}(\mathbb{1}_{\lfloor i \div 3 \rfloor}) + \mathbf{W}_{\text{val}}(\mathbb{1}_{v_i})\}_{i=0}^{t-1},$$

where $\mathbb{1}$ maps its integer argument in the subscript into a one-hot vector, $\mathbf{W}_{\text{coo}} \in \mathbb{R}^{d_{\text{emb}} \times 3}$, $\mathbf{W}_{\text{pos}} \in \mathbb{R}^{d_{\text{emb}} \times |\mathcal{V}|}$ and $\mathbf{W}_{\text{val}} \in \mathbb{R}^{d_{\text{emb}} \times 2^6}$ are all learned matrices that map their inputs to $d_{\text{emb}}=256$ dimensional embeddings. Then a Transformer decoder $T^{\mathcal{V}}_{\text{dec}}$ model predicts a probability distribution (in the form of logits) over all possible vertex locations for the next token $v_t$:

$$p(v_t) = T^{\mathcal{V}}_{\text{dec}}(\mathbf{h}_{\mathcal{V}^{\text{seq}}_{<t}}).$$

The model is trained using teacher-forcing (Williams & Zipser, 1989) to maximize the log-likelihood of the training data with a cross-entropy loss with label smoothing of 0.01.

## 4.2 Edge Model

The edge model (Figure 4, center) learns a distribution $p(\mathcal{E})$ over the vertex indices. The edges are represented as a 1D list $\mathcal{E}^{\text{seq}}$ by flattening the vertex indices in $\mathcal{E}$, with a new edge token <NEW_EDGE> marking the start

of a new edge and a stopping token `<EOS>` marking the end of the list. For ordering invariance, we sort each edge $E \in \mathcal{E}$ in ascending order and then sort $\mathcal{E}$ such that the edges with lowest vertex indices come first. The edge sequence $\mathcal{E}^{\text{seq}} = \{e_0, e_1, \texttt{<NEW\_EDGE>}, \ldots, \texttt{<EOS>}\}$ is of length $|\mathcal{E}^{\text{seq}}| = \sum_{E \in \mathcal{E}} |E| + 1$, and the edge model learns the following distribution:

$$p(\mathcal{E}^{\text{seq}}|\mathcal{V}; \theta_{\mathcal{E}}) = \prod_{j=0}^{|\mathcal{E}^{\text{seq}}|-1} p(e_j \mid e_0, e_1, \ldots, e_{j-1}, \mathcal{V}; \theta_{\mathcal{E}}), \tag{5}$$

where $\theta_{\mathcal{E}}$ are learnable parameters of the neural network.

Given the input vertices $\mathcal{V}$ and current edges $\mathcal{E}^{\text{seq}}_{<t}$ until the step $t$, the goal is to model the next output vertex index $e_t$ as a probability distribution over the indices of $\mathcal{V}$. This is done by first learning *value embeddings* $\mathbf{h}_{\mathcal{V}} \in \mathbb{R}^{|\mathcal{V}| \times d_{\text{emb}}}$ from the input vertices $\mathcal{V} = \{(x_i, y_i, z_i)\}_{i=0}^{|\mathcal{V}|-1}$:

$$\mathbf{h}_{\mathcal{V}} = \{\phi\left(\mathbf{W}_{\text{x}} \mathbb{1}_{x_i} \parallel \mathbf{W}_{\text{y}} \mathbb{1}_{y_i} \parallel \mathbf{W}_{\text{z}} \mathbb{1}_{z_i}\right)\}_{i=0}^{|\mathcal{V}|-1}, \tag{6}$$

where $\mathbf{W}_{\text{x}}$, $\mathbf{W}_{\text{y}}$ and $\mathbf{W}_{\text{z}} \in \mathbb{R}^{64 \times 2^6}$ are learned matrices that map their inputs into 64-dimensional embeddings, $\parallel$ is the concatenation operation along the second dimension, and $\phi \in \mathbb{R}^{d_{\text{emb}} \times (64 \times 3)}$ is a learned linear layer to produce $d_{\text{emb}}=256$ dimensional embeddings. The token embeddings for `<NEW_EDGE>` and `<EOS>`, $\mathbf{h}_{\texttt{<NEW\_EDGE>}} \in \mathbb{R}^{d_{\text{emb}}}$ and $\mathbf{h}_{\texttt{<EOS>}} \in \mathbb{R}^{d_{\text{emb}}}$ are learnable parameters that are concatenated to form a total of $|\mathcal{V}| + 2$, which is further processed by Transformer encoder $T^{\mathcal{E}}_{\text{enc}}$ to obtain $\mathbf{h}_{\text{inp}} \in \mathbb{R}^{(d_{\text{emb}} \times |\mathcal{V}|+2)}$

$$\mathbf{h}_{\text{inp}} = T^{\mathcal{E}}_{\text{enc}}(\mathbf{h}_{\texttt{<EOS>}} \parallel \mathbf{h}_{\texttt{<NEW\_EDGE>}} \parallel \mathbf{h}_{\mathcal{V}}).$$

Then, edge embeddings $\mathbf{h}_{\mathcal{E}^{\text{seq}}_{<t}} \in \mathbb{R}^{|\mathcal{E}^{\text{seq}}_{<t}| \times 256}$ are formed by gathering the input embeddings $\mathbf{h}_{\text{inp}}$ corresponding to the vertex indices in $\mathcal{E}^{\text{seq}}_{<t}$ and summed with learned *positional embeddings* indicating the position of each token in $\mathcal{E}^{\text{seq}}_{<t}$.

$$\mathbf{h}_{\mathcal{E}^{\text{seq}}_{<t}} = \{\mathbf{h}_{\text{inp}}[e_j] + \mathbf{W}_{\text{pos}} \mathbb{1}_j\}_{j=0}^{t-1},$$

where $\mathbf{W}_{\text{pos}} \in \mathbb{R}^{|\mathcal{E}^{\text{seq}}| \times d_{\text{emb}}}$ is a learnable matrix.

A Transformer decoder $T^{\mathcal{E}}_{\text{dec}}$ then processes these embeddings followed by a linear layer to output a pointer vector $\mathbf{p}_t$ that is compared to the input embeddings $\mathbf{h}_{\text{inp}}$ using a dot product operation and normalized using softmax to get a distribution over the $0 \leq k \leq |\mathcal{V}| + 1$ indices including the vertex indices and the `<NEW_EDGE>`, `<EOS>` tokens:

$$\mathbf{p}_t = T^{\mathcal{E}}_{\text{dec}}(\mathbf{h}_{\mathcal{E}^{\text{seq}}_{<t}}),$$
$$p(e_t = k \mid \mathcal{E}^{\text{seq}}_{<t}, \mathcal{V}) = \text{softmax}_k(\mathbf{p}_t \cdot \mathbf{h}_{\text{inp}}[k]).$$

The model is trained using the cross-entropy loss to predict the distribution $p(e_t = k \mid \mathcal{E}^{\text{seq}}_{<t}, \mathcal{V})$ which is repeatedly created at each time step and sampled to generate the edge tokens autoregressively. During training, ground-truth is used for conditioning (teacher-forcing) rather than previous samples.

### 4.3 Face Model

The face model (Figure 4, right) learns a distribution $p(\mathcal{F})$ over the faces. The faces are represented as a flat sequence $\mathcal{F}^{\text{seq}}$ similar to the edges by flattening $\mathcal{F}$ into a 1D list of edge indices, with a new face token `<NEW_FACE>` marking the start of a new face and a stopping token `<EOS>` marking the end of the faces list. The model learns the following probability distribution:

$$p(\mathcal{F}^{\text{seq}}|\mathcal{E}, \mathcal{V}; \theta_{\mathcal{F}}) = \prod_{t=0}^{|\mathcal{F}^{\text{seq}}|-1} p(f_t \mid f_0, f_1, \ldots, f_{t-1}|\mathcal{E}, \mathcal{V}; \theta_{\mathcal{F}}), \tag{7}$$

where $\theta_{\mathcal{F}}$ are learnable parameters of the neural network.

The face model functions similarly as the edge model and represents the face features by gathering edge embeddings. Given the input vertices $\mathcal{V}$ and edges $\mathcal{E}$, and the current faces $\mathcal{F}^{\text{seq}}_{<t}$ at step $t$, the goal is to model the next output face token $f_t$ as a probability distribution over the indices of $\mathcal{E}$.

This is done by learning vertex embeddings $\mathbf{h}'_{\mathcal{V}} \in \mathbb{R}^{|\mathcal{V}| \times d_{\mathrm{emb}}}$ from the input vertices as in Equation 6. The rows of $\mathbf{h}'_{\mathcal{V}}$ corresponding to the vertex indices in $\mathcal{E}$ are first gathered and summed, and learned embeddings for `<NEW_FACE>` and `<EOS>`, $\mathbf{h}_{\texttt{<NEW\_FACE>}}$ and $\mathbf{h}_{\texttt{<EOS>}}$ are further concatenated to these embeddings, and passed through a Transformer encoder $\bar{T}^{\mathcal{E}}_{\mathrm{enc}}$ to form $\mathbf{h}_{\mathcal{E}} \in \mathbb{R}^{(|\mathcal{E}|+2) \times d_{\mathrm{emb}}}$:

$$\mathbf{h}_{\mathcal{E}} = \bar{T}^{\mathcal{E}}_{\mathrm{enc}}(\{\mathbf{h}_{\texttt{<EOS>}} \,\|\, \mathbf{h}_{\texttt{<NEW\_FACE>}} \,\|\, \{\textstyle\sum_{v \in E} \mathbf{h}'_{\mathcal{V}}[v])\}_{E \in \mathcal{E}}),$$

Finally, face embeddings $\mathbf{h}_{\mathcal{F}^{\mathrm{seq}}_{<t}}$ are formed by gathering the edge embeddings which is further summed with learned *positional embeddings*, and passed through another Transformer encoder $T^{\mathcal{F}}_{\mathrm{enc}}$:

$$\mathbf{h}_{\mathcal{F}^{\mathrm{seq}}_{<t}} = T^{\mathcal{F}}_{\mathrm{enc}}(\{\mathbf{h}_{\mathcal{E}}[f_k] + \mathbf{W}'_{\mathrm{pos}} \mathbb{1}_k\}_{k=0}^{t-1}),$$

where $\mathbf{W}'_{\mathrm{pos}} \in \mathbb{R}^{|\mathcal{F}^{\mathrm{seq}}| \times d_{\mathrm{emb}}}$ is a learnable matrix. The idea here is to encode faces by the embeddings of the edges that lie on their boundary, and encode the edges by the embeddings of the incident vertices.

A Transformer decoder $T^{\mathcal{F}}_{\mathrm{dec}}$ then processes $\mathbf{h}_{\mathcal{F}^{\mathrm{seq}}_{<t}}$ to output a pointer vector $\mathbf{p}_t$ that is compared to the edge embeddings $\mathbf{h}_{\mathcal{E}}$ using a dot product operation and normalized using softmax to get a distribution over the $0 \leq k \leq |\mathcal{E}| + 1$ input edge indices and `<NEW_FACE>`, `<EOS>` tokens:

$$\mathbf{p}_t = T^{\mathcal{F}}_{\mathrm{dec}}(\mathbf{h}_{\mathcal{F}^{\mathrm{seq}}_{<t}}),$$
$$p(f_t = k \mid \mathcal{F}^{\mathrm{seq}}_{<t}, \mathcal{E}, \mathcal{V}) = \mathrm{softmax}_k(\mathbf{p}_t \cdot \mathbf{h}_{\mathcal{E}}[k]).$$

The face model is also trained by teacher-forcing using a cross-entropy loss.

## 5 Experiments

In this section we perform experiments to qualitatively and quantitatively evaluate our method on unconditional generation, and various conditional generation tasks based on class labels, images, and voxels.

**Implementation.** Our implementation is in PyTorch (Paszke et al., 2019). We train our models for 1000 epochs with batch size 512 using the AdamW optimizer (Loshchilov & Hutter, 2019) (learning rate: $10^{-4}$, weight decay: 0.01) on an Nvidia DGX A100 machine. The vertex, edge, and face models can be trained independently since we employ teacher-forcing where the models are decoupled during training—rather than using the samples from one model to condition the subsequent model, we use the ground truth data. When a conditional embedding is jointly learned however, we have to train the three models together. Training time ranges from 1–3 days depending on the dataset. We find it critical for convergence to use the pre-LayerNorm variant of the Transformer (Xiong et al., 2020). All Transformer modules use 8 layers with an embedding dimension of 256, and fully-connected dimension of 512 and 8 attention heads. We initially experimented with 4, 8 and 12-layer models and observed the 8-layer model to work well. The 4-layer model was slightly underfitting while the 12-layer model was overfitting on our datasets. To sample from the models we use nucleus sampling (Holtzman et al., 2020). We find it helpful to mask logits that are invalid in each step of the sampling (see Section A.3). All data processing related to building indexed B-reps and reconstructing B-reps uses the OpenCascade/pythonOCC (Paviot, 2008) solid modeling kernel.

**Datasets.** Publicly available datasets for B-rep solid models are limited compared to other 3D representations. We demonstrate our method on two datasets while considering only files that contain the surface and curve types supported by the indexed B-rep format.
*The Parametric Variations (PVar) Dataset* is synthetically designed (see Section A.5 for details) for testing SolidGen on the class-conditional generation task, since categorically labeled B-rep datasets are unavailable. There are 120,000 models, 2000 in each of the 60 classes. The dataset is split in a 90 (train)/5 (validation)/5 (test) proportion.
*The DeepCAD Dataset* (Wu et al., 2021) contains a subset of the CAD models available in the ABC dataset (Koch et al., 2019) that additionally includes the sequence of sketch and extrude CAD modeling operations. The original dataset contains 178,238 CAD models with a significant portion of duplicate and trivial models. We use a hash-based method, described in Section A.1, to remove duplicates. We filter out

Table 1: Modeling metrics for unconditional and conditional SolidGen models computed on the test set.

| Dataset | Model | NLL (bits per, ↓) | | | | Top-1 Accuracy (%, ↑) | | | |
|---------|-------|------|------|------|-------|-------|------|------|------|
| | | Vert. | Edge | Face | Total | Vert. | Edge | Face | Mean |
| PVar | Uniform | 18.44 | 13.59 | 24.96 | 56.99 | 2.01 | 1.59 | 2.04 | 1.88 |
| | SolidGen | 4.49 | 0.01 | 0.04 | 4.54 | 91.30 | 99.97 | 99.88 | 97.05 |
| | w/class (vertex) | 4.19 | - | - | 4.24 | 83.20 | - | - | 94.35 |
| | w/class (all) | 1.28 | 0.01 | 0.03 | 1.32 | 92.43 | 99.97 | 99.89 | 97.43 |
| DeepCAD | Uniform | 18.32 | 15.05 | 28.14 | 61.51 | 1.48 | 1.10 | 1.42 | 1.33 |
| | SolidGen | 5.42 | 0.43 | 0.26 | 6.11 | 86.12 | 98.87 | 99.55 | 94.85 |
| | w/image (vertex) | 1.97 | - | - | 2.66 | 80.61 | - | - | 93.01 |
| | w/image (all) | 1.98 | 0.26 | 0.27 | 2.51 | 89.74 | 98.84 | 99.31 | 95.96 |
| | w/voxel (vertex) | 4.89 | - | - | 5.42 | 82.28 | - | - | 93.48 |
| | w/voxel (all) | 2.00 | 0.52 | 0.65 | 3.17 | 91.26 | 98.62 | 99.18 | 96.35 |

trivial ($< 8$ faces, for e.g., boxes), overly-complex ($> 130$ faces) models, those that contain merged vertices after quantization and ones that yield very long sequences with $> 200$ tokens. We are left with 49,759 models that are split in 90/5/5 train/validation/test sets and used for all experiments. For image conditioning, we render images of the CAD models with the OpenCascade viewer. For voxel conditioning, we sample a point cloud with 2048 points from the surface of the CAD models, and quantize them into voxel grids of dimension $(2^6)^3$ matching our vertex quantization.

**Metrics.** We use the following metrics to quantitatively evaluate and compare SolidGen with other methods. *Valid* reports the percentage of solids that were successfully built and considered valid by the solid modeling kernel (see Section A.6). *Novel* measures the percentage of valid solids that are not duplicated from the training set, and *Unique* reports the percentage of valid solids not duplicated within the sample set. The duplication check is based on our hashing procedure described in Section A.1.

## 5.1 Unconditional Generation

In this section we evaluate SolidGen on unconditional generation of B-reps.

**Modeling Performance.** To evaluate the modeling performance, we report the negative log-likelihood (NLL) and per-step accuracy computed on the DeepCAD test set. The NLL is reported individually for our vertex, edge and face models in bits per vertex, bits per edge and bits per face, respectively. Since there is no other method that uses our indexed B-rep format, a one-to-one comparison is not possible and we use a 'Uniform' model that allocates uniform probability to the entire co-domain of the data as a baseline reference. The accuracy corresponds to the top-1 prediction correctness of the next token given the ground truth for the previous tokens. Table 1 (first two rows under 'DeepCAD') shows the quantitative metrics of our model and the uniform baseline on the DeepCAD (Wu et al., 2021) dataset. SolidGen obtains an NLL of 4.54 bits and 97.05% accuracy compared to the 'Uniform' model that has an NLL of 56.99 bits and 1.88% accuracy. We find that the edge and face models always outperform the vertex model. This could be because once the vertices are estimated, the edges and faces are significantly constrained in our representation. We show plots for top-10 accuracy in Section A.8.

**Comparison with DeepCAD.** We evaluate the quality of the generated samples produced by SolidGen and compare them with that of DeepCAD (Wu et al., 2021) model. Note that both models work in fundamentally different ways—SolidGen is an autoregressive generative model trained directly on the B-reps, while DeepCAD is an autoencoder that learns from sequences of CAD modeling operations. This fundamental difference in representation leads to differences in design choices making a fair comparison challenging. We strive to provide a one-to-one comparison by focusing mainly on the sample quality. Figure 5 shows qualitative results for unconditional generation trained on the DeepCAD dataset Wu et al. (2021). We show samples from the dataset for comparison (a), along with generated samples from DeepCAD (b) and SolidGen (c). More results are shown in Section A.8. We present quantitative metrics in Table 2. A large portion

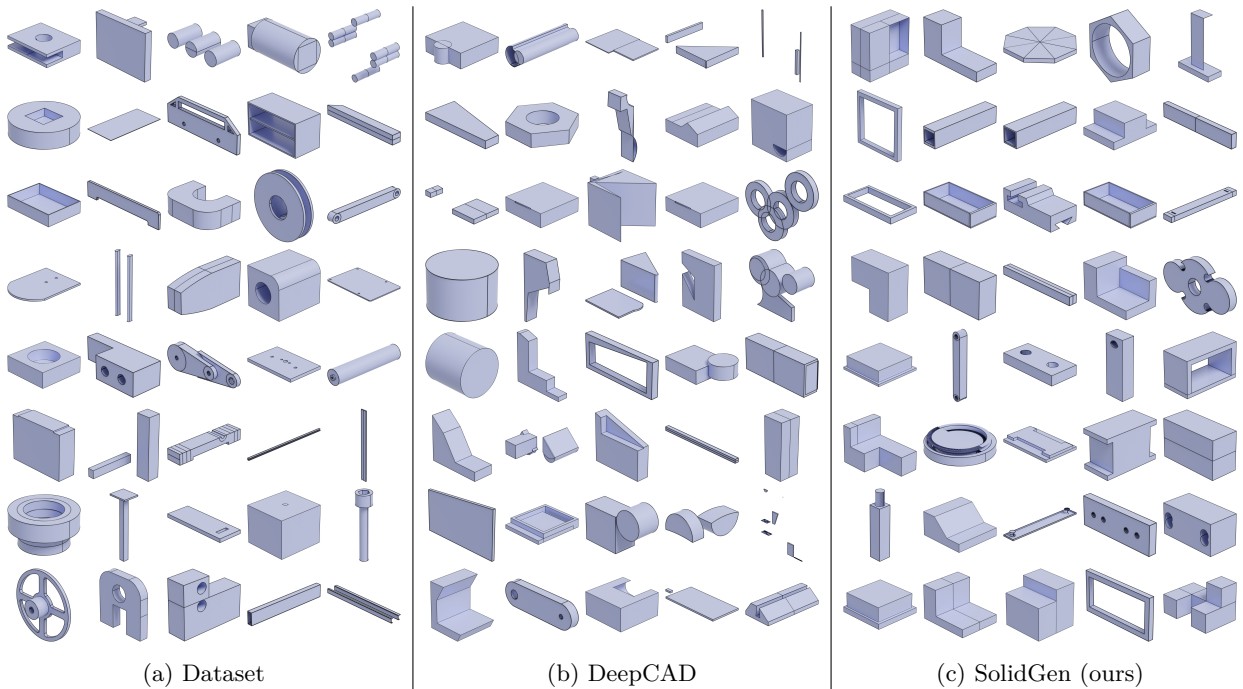

|              (a) Dataset              |              (b) DeepCAD              |              (c) SolidGen (ours)              |

Figure 5: Qualitative results for unconditional generation. Samples (a) from the DeepCAD dataset, (b) generated by the DeepCAD model (Wu et al., 2021), and (c) generated by SolidGen.

Table 2: Quality of 5000 unconditional samples generated by networks trained on the DeepCAD dataset. SolidGen results are shown for samples generated with different top-$p$ values in nucleus sampling.

| Model | Valid (%,↑) | Novel (%,↑) | Unique (%,↑) |
|---|---|---|---|
| SolidGen (p=0.5) | 87.57 | 66.22 | 33.69 |
| SolidGen (p=0.6) | 88.12 | 73.00 | 61.42 |
| SolidGen (p=0.7) | 86.70 | 82.46 | 83.91 |
| SolidGen (p=0.8) | 85.21 | 88.09 | 93.65 |
| SolidGen (p=0.9) | 83.10 | 92.38 | 97.49 |
| DeepCAD | 62.11 | 96.73 | 99.01 |

of the models produced by SolidGen are valid and usable B-reps, with SolidGen showing between 20.99% to 26.01% improvement in the valid ratio compared to DeepCAD. By varying the top-$p$ hyperparameter in nucleus sampling, SolidGen is able to trade-off between validity and novelty and uniqueness of B-reps produced. We observe a clear trend where increasing $p$ reduces validity while increasing the novelty and uniqueness. SolidGen falls slightly below DeepCAD and obtains 4.38% to 30.51% lower novel and 1.52% to 65.32% lower unique ratio depending on the top-$p$ parameter. We found that many of DeepCAD's samples contain self-intersections and tend to be noisy/unrealistic leading to decreasing the valid score, but artificially increasing the unique and novel scores. We further investigate this through a human perceptual study next.

**Human Perceptual Evaluation.** Rather than rely only on metrics, we believe it is critical to perform human evaluation on the output of generative models. To evaluate the realism of the generated samples, we perform a perceptual study using human evaluators recruited through Amazon's Mechanical Turk service (Mishra, 2019). The crowd workers were shown pairs of images, one of which was generated by SolidGen ($p = 0.7$) or DeepCAD and the other randomly selected from the training set, and asked to judge which of the two was more realistic. Each image pair was rated by 7 different human evaluators and we record the number of crowd workers who selected the generated model as more realistic. The results are shown in Figure 6. We see that for SolidGen the distribution is roughly symmetric with the majority vote judging the

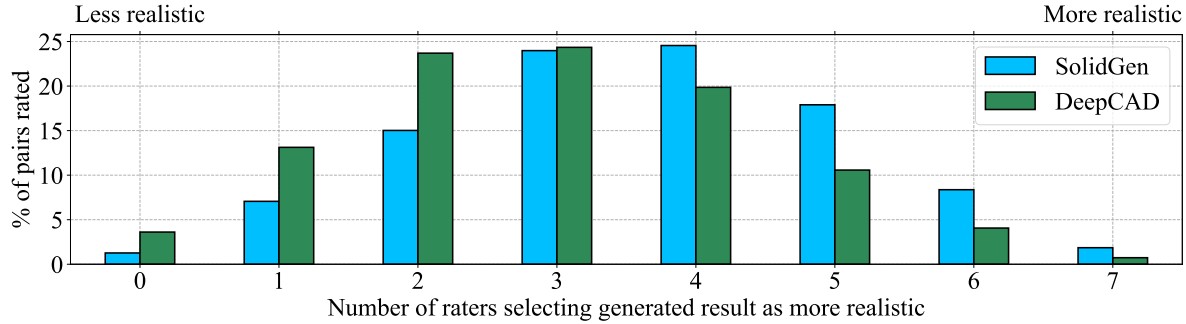

Figure 6: Human perceptual study showing the distribution of votes by 7 human evaluators for the realism of SolidGen and DeepCAD models compared to samples from the training set.

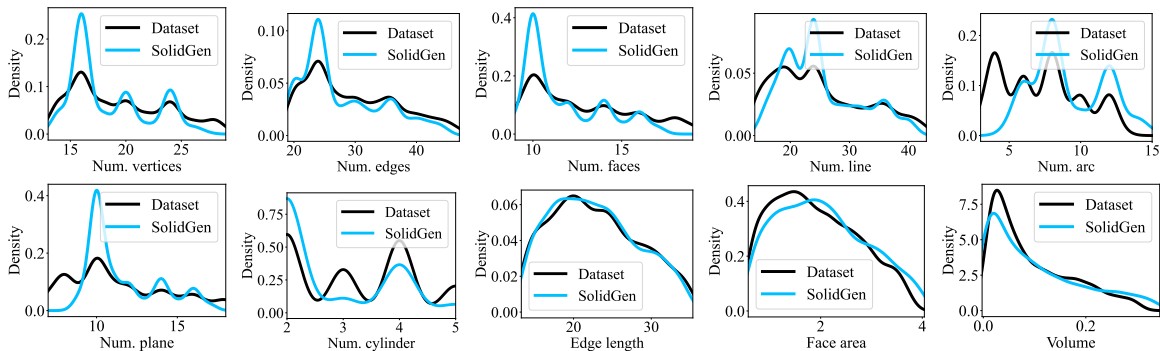

Figure 7: Comparison of distributions computed from SolidGen samples and the DeepCAD training set.

SolidGen output to be more realistic 52.67% of the time. This indicates the realism of the SolidGen output is indistinguishable from the training set. For DeepCAD we see the distribution is skewed to the left, with the majority vote judging the DeepCAD output to be more realistic only 35.22% of the time.

**Statistics.** To check if SolidGen synthesizes B-reps representative of the dataset, we compute several statistics between the samples produced by SolidGen (with top-$p$=0.9) compared to the DeepCAD training set in Figure 7. Here we used kernel density estimation with Gaussian kernels to fit the curves, and see that SolidGen well captures the modes of the data distribution.

## 5.2 Conditional Generation

In this section, we demonstrate various conditional generation results using class labels in the PVar dataset and rendered images and voxels from the DeepCAD dataset. Conditional models use the same architecture, hyperparameters and training strategy as the unconditional model, and support two kinds of conditioning:
*Start-of-sequence (SOS) Embedding.* For class conditioning, we learn a $d_{\text{emb}}$=256 dimensional embedding from the input class label, and use it as the SOS embedding in the vertex, edge and face models.
*Cross Attention.* For image/voxel conditioning, we use a 2D/3D convolutional neural network (details in Section A.7) to obtain an $16^2/8^3$ dimensional embedding that is flattened and jointly used by the Transformer decoders in the vertex, edge and face models via cross-attention.

**Modeling Performance.** The modeling performance of the conditional models are shown in Table 1. The 'w/class (all)' row under 'PVar' shows that class conditioning improves the NLL by $-3.22$ and accuracy by $+0.38\%$ compared to the unconditional 'SolidGen' model trained on the same data. A similar trend is observed for image and voxel conditioning shown under row 'DeepCAD'. The NLL for the 'w/image (all)' conditional model improved by $-3.6$ while the accuracy improved by $+3.6\%$. For the voxel conditional model 'w/voxel (all)', the NLL improved by $-2.94$ while the accuracy increased by $+1.5\%$.

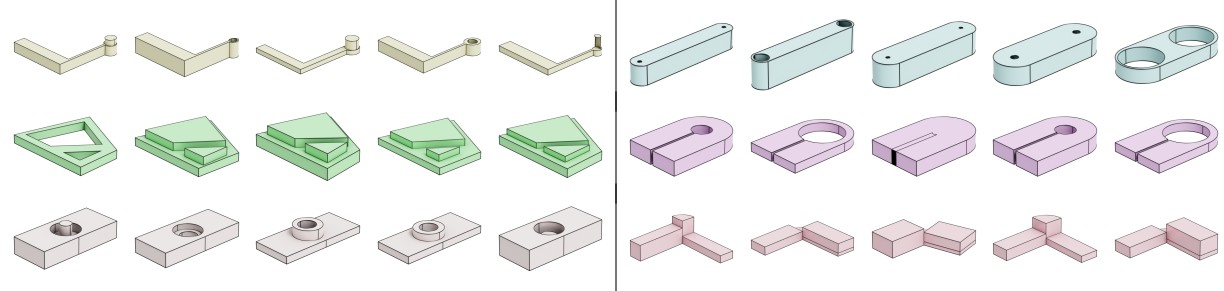

Figure 8: Class conditional samples (each row is a unique class) from SolidGen trained on the PVar dataset.

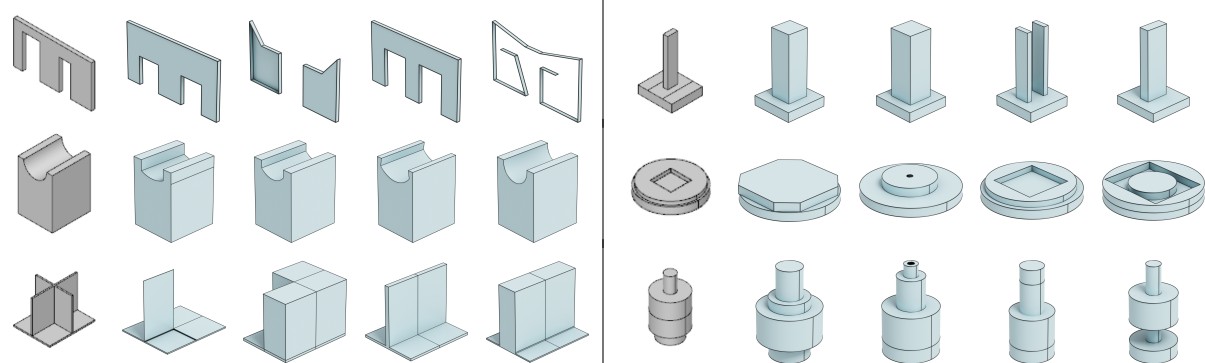

Figure 9: Image conditional samples from SolidGen using images (first column) obtained by rendering CAD models in the DeepCAD test set.

**Sample Quality.** Qualitative results of our class-conditioning model given random class labels as input are shown in Figure 8. Figure 9 shows B-reps produced by our model given images as input conditioning, while Figure 10 shows results where voxelized point clouds were used as conditioning. We see that SolidGen is able to synthesize design variations that are close to the input conditioning. Quantitative evaluation of the class-conditional samples (40 per-class), image conditional samples (20 per image) and voxel conditional samples (20 per voxel grid) are provided in Table 3. We see that SolidGen generates a high number of novel B-reps, but the valid and unique ratio go down as we transition from class to image to voxel conditioning. We suspect this trend is due to data imbalance in the dataset which contains arbitrary shapes without clear categories making the overall task challenging. Moreover, unlike unconditional and class conditional generation, image and voxel conditioning require generating design variations for every data in the test set,

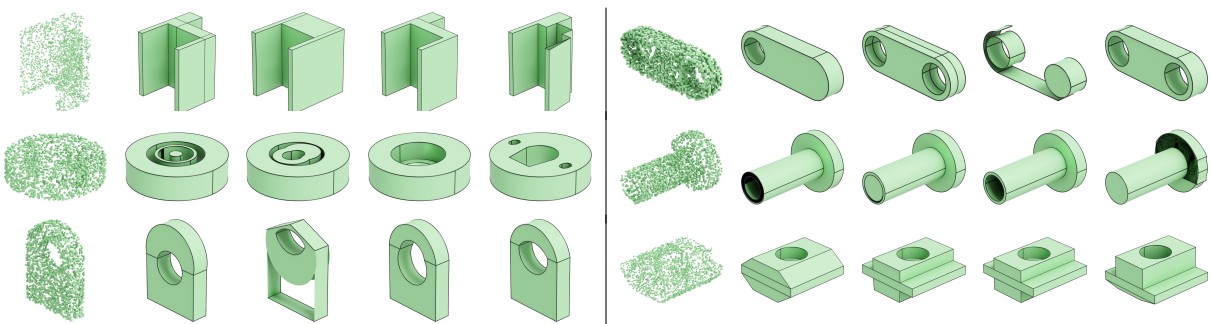

Figure 10: Voxel conditional samples from SolidGen using voxelized point clouds (first column) obtained by sampling 2048 points from CAD models in the DeepCAD test set.

Table 3: Quality of conditional samples generated by SolidGen with top-$p$=0.7.

| Conditioning | Valid (%,↑) | Novel (%,↑) | Unique (%,↑) |
|---|---|---|---|
| Class (PVar) | 83.48 | 92.35 | 70.71 |
| Image (DeepCAD) | 71.45 | 94.61 | 75.08 |
| Voxel (DeepCAD) | 66.78 | 92.75 | 53.95 |

making the task more challenging. The less precise nature of 2D image conditioning makes it easier to learn and generate design variations compared to the 3D case.

**Ablation on joint conditioning.** Conditional models require the vertex, edge and face models to be trained together. To understand the importance of jointly learning a conditional embedding for the vertex, edge and face models, we experimented with a variant of SolidGen where the conditioning is only applied to the vertex model. This variant denoted by 'w/ $*$(vertex)' in Table 1 has the advantage that the vertex, edge, and face models can be trained independently in parallel since the probability distribution being modeled is $p(\mathcal{B}|z) = p(\mathcal{F}|\mathcal{E}, \mathcal{V})p(\mathcal{E}|\mathcal{V})p(\mathcal{V}|z)$. We observe that conditioning only the vertex model does improve the NLL in all cases (class:$-0.3$, image:$-3.45$, voxel:$-0.69$), but the accuracy drops below that of the unconditional model (class:$-2.7\%$, image:$-1.84\%$, voxel:$-1.37\%$). The joint conditioning variants 'w/class (all)', 'w/image (all)' and 'w/voxel (all)' perform the best in all cases.

## 6    Conclusion

**Limitations & Future Work.** Our current method has some limitations that can be addressed in future work. Extremely complex CAD models with long sequence lengths increase the training time and chance for compounding errors at test time, which could be attributed to teacher forcing. Scaling our model and training on larger CAD datasets might alleviate the problem. Our indexed B-rep format supports the most common curve and surface types found in mechanical CAD models but not conic sections and splines that are common in freeform modeling. Uniform B-spline curves of a fixed degree can be supported by considering edges that group $>= (\text{degree} + 1)$ vertices (the spline's control points). However, unlike prismatic surfaces, the B-spline surfaces cannot be fully determined by the boundary curves, and require the additional prediction of a grid of interpolating or control points. Like previous generative neural networks (Willis et al., 2021b; Nash et al., 2020; Wu et al., 2021; Xu et al., 2022) our method is trained using classification losses. Several important CAD applications, e.g., reverse engineering, require reconstruction losses that incorporate the B-rep geometry which is not available until postprocess. Finally, research into conditioning schemes could facilitate stronger user guidance in the generation, and latent representations would help with better generalization.

**Summary.** We presented SolidGen, a generative model that can directly learn from and synthesize boundary representation (B-rep) CAD models without the need for supervision from a sequence of CAD modeling operations. We achieved this by deriving the indexed B-rep format that captures the hierarchical nature of B-rep vertices, edges and faces in a new machine learning friendly representation. SolidGen generates highly coherent yet diverse B-reps as demonstrated in our comparison with prior work. Our method has potential to be integrated into CAD software workflows, since all CAD software allows the import of solids without modeling history. As our method can generate local regions of B-rep topology, in addition to entire solids, this allows learning-based techniques to play a role in many other workflows such as solid model inpainting and parting surface creation. Conditional generation can aid in sketch-based modeling workflows and to convert point clouds, meshes or other file formats into B-reps for further editing.

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

# A    Appendix

## A.1    Hashing Indexed B-reps

We use duplication detection (Willis et al., 2021b) tools to compare B-reps. This is used, for example, to assess the degree to which the generated B-reps exactly match with those in the training set, quantify the diversity of samples generated by SolidGen, etc. The duplicate detection algorithm is based on the Weisfeiler Lehman isomorphism test. Two graphs are built from the B-rep—a face adjacency graph (Ansaldi et al., 1985) where the B-rep faces are graph nodes and adjacent faces are connected by graph edges, and a vertex adjacency graph where B-rep vertices are graph nodes and the B-rep edges are graph edges. The quantized vertex positions are used to initialize the vertex hashes and stored as node attributes, and the curve type information is stored as edge attributes, making the final hash dependent on the solid geometry. The final hash string for each solid is created by concatenating the graph hashes of the face adjacency and vertex adjacency graphs.

## A.2    Converting Indexed B-reps to B-reps

Here we detail the complete method to convert indexed B-reps $\mathcal{B} = \{\mathcal{V}, \mathcal{E}, \mathcal{F}\}$ to B-reps.

### A.2.1    Vertices and Edges

Each point in $\mathcal{V}$ is potentially a B-rep vertex. Each edge $E \in \mathcal{E}$ is used to dereference into $\mathcal{V}$ and collect the list of points defining the edge and its geometry. The endpoints of $E$ are used to construct a pair of B-rep vertices if they do not already exist, and the topological B-rep edge connecting them. The curve geometry is defined by the cardinality of $E$: two points define a line, while three points define an arc (see Figure 3 right). Since we sort the indices in each edge in ascending order for ordering invariance, the start point, arc midpoint and end point must be identified. We do this by considering the permutations of the three points, and choose the one where the second point coincides with the point evaluated at the middle parameter of the arc.

### A.2.2    Wires

Next, we examine each face in $\mathcal{F}$ to gather the list of edges bounding it. Note that edges bounding a face can either be part of an outer wire running in counter-clockwise direction that defines the visible part of the face's geometry, or part of one or more inner wires running in clockwise direction that define the hidden portion (holes) in the face's geometry. Our goal here is to form these wires by connecting the given set of edges into one or more closed loops. We achieve this by defining a vertex-edge graph where each B-rep edge and its vertices are graph nodes connected by directed edges and detecting cycles in this graph (see Figure A1 (a)). Each cycle (except trivial ones) defines a wire, and the outer wire is defined as the largest wire (based on the extents of its bounding box). The other wires are assumed to run in clockwise direction and define holes.

### A.2.3    Surfaces

We support five prismatic parametric surface types that are common in CAD: plane, cylinder, cone, sphere, and torus. We develop an algorithm that identifies the surface type by finding the simplest surface consistent with the boundary curves in the wires bounding a face within a prescribed tolerance (see Figure A1 (b)).

If all curves in the face are coplanar then a plane is fitted. Otherwise, the curves are checked for consistency with a cylinder, cone, sphere and torus in that order.

If a face is trimmed by a combination of lines and arcs then the surface will either be a cylinder or a cone. For a cylinder the radii of the arcs must be the same, the normals must be parallel and the centers aligned in the normal direction. Any lines must also be parallel to the normal direction. If these conditions are not met then the surface is treated as a cone.

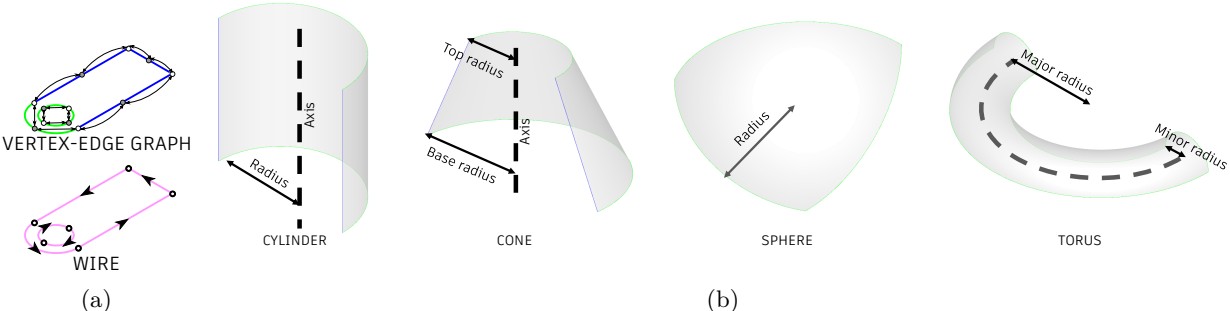

Figure A1: (a) Vertex-edge graph used to build and orient the wires by finding cycles. (b) Identifying different kinds of surfaces from the curves (lines in blue, arcs in green) in the outer wire.

If a face contains only arcs, we can define a sphere whose center lies at the center of an arc, and check for consistency with the rest of the arcs. If a sphere is not an appropriate fit, we define a torus with the major radius set to the average of the two largest arcs and minor radius set to that of the smallest arc.

The only ambiguity with this procedure is when a face contains a single circle on its boundary—the surface could be a plane trimmed into a disk or a hemisphere. Real world CAD data does not contain many examples like this. Spheres typically appear only on corner fillets as part of three-sided faces where the presence of three non-coplanar arcs clearly disambiguates this case.

### A.2.4 Faces

With wires and surfaces available, it is straightforward to combine them to build a face using the solid modeling kernel. Finally, to topologically connect the individual faces with each other, we sew the faces into shells, and further attempt to stitch the shells into a single solid model.

### A.3 Masking Invalid Logits

During the autoregressive sampling, we mask the invalid logits (by setting their values to a small number e.g. $-10^9$) that do not satisfy these criteria at each time step, and distribute the next token probabilities among the valid logits as in Nash et al. (2020) (which happens automatically when applying the softmax operation). Let $t$ denote the current step in the sampling starting from $t = 0$ for the first step.

For the vertex model:

- If the vertex token being generated is a z-coordinate ($t \mod 3 = 0$) then it has to be greater than or equal to the previous z-coordinate that was generated.

- If the vertex token being generated is a y-coordinate ($t \mod 3 = 1$), and the last two z-coordinates were equal, then it has to be greater than or equal to the previous y-coordinate that was generated.

- If the vertex token being generated is a x-coordinate ($t \mod 3 = 2$), and the last two z-coordinates and y-coordinates were equal, then it has to be greater than the previous x-coordinate that was generated.

- The <EOS> token can only appear after an x-coordinate i.e. when $t > 0$ and $t \mod 3 = 0$.

For the edge model:

- The <EOS> and <NEW_EDGE> tokens can only appear if $t \neq 0$ and cannot be repeated consequently.

- If the previous token was a <NEW_EDGE>, then the current token has to be greater than or equal to first token in the previous edge to respect the sorted ordering of the edges.

- If the previous token was not a <NEW_EDGE>, then the current token has to be greater than the previous token to respect the sorted order of tokens within each edge.

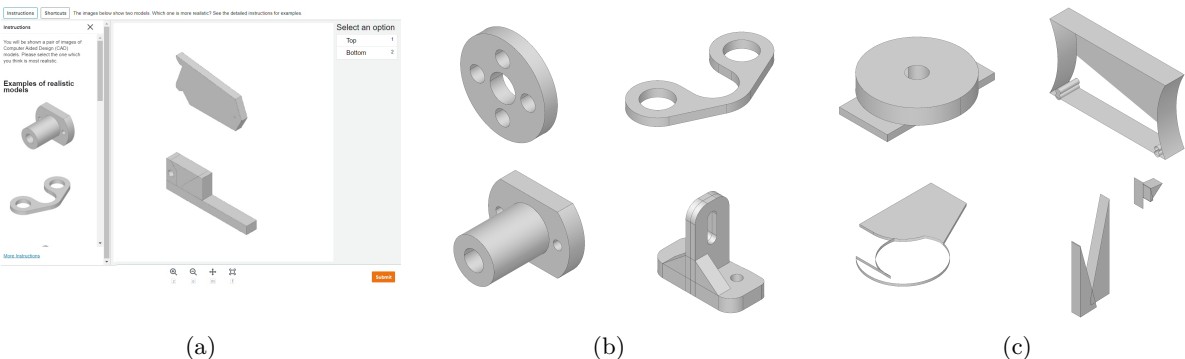

Figure A2: (a) An example of an image pair shown to the crowd workers. (b) Examples of "realistic" models (from the DeepCAD training set) shown to the crowd workers. (c) Examples of unrealistic models. Two of these were generated by DeepCAD and two by SolidGen.

- A `<NEW_EDGE>` or `<EOS>` tokens can only appear after two or three edge tokens have been generated. This guarantees that we define an edge as a line or an arc.

For the face model:

- The `<EOS>` and `<NEW_EDGE>` tokens can only appear if $t \neq 0$ and cannot be repeated consequently.

- If the previous token was a `<NEW_FACE>` token, then the current token has to be greater than or equal to first token in the previous face to respect the sorted ordering of the faces.

- If the previous token was not a `<NEW_FACE>`, then the current token has to be greater than the previous token to respect the sorted order of tokens within each face.

- A `<NEW_FACE>` or `<EOS>` tokens can only appear at least two face tokens have been generated. This guarantees that each face at least two edges to make it closed e.g. two arcs.

- An edge index can only be used twice in the sampled face tokens. This ensures that an edge can at most be shared by two faces, and helps prevents non-manifold results.

### A.4 Human Evaluation

As discussed in Section 5.1, to evaluate the realism of the models generated by SolidGen, we perform a perceptual study using human evaluators recruited through Amazon's Mechanical Turk service (Mishra, 2019). The crowd workers were shown pairs of images, one of which was generated by SolidGen or DeepCAD and the other randomly selected from the training set. The position of the image of the generated model was randomized to be at the top or the bottom of the pair. An example of an image pair is shown in Figure A2a. The crowd workers were asked to select the model which they found to be most "realistic". To assist with this task we provided four examples of realistic models (Figure A2b) and "unrealistic" models (Figure A2c). The realistic examples were carefully chosen from the training set to include desirable properties like symmetry and clear design intent and function. For the unrealistic examples we deliberately selected models which had obvious problems like missing faces or which formed an incoherent collection of shapes.

In total 4900 pairs of images were shown to the crowd workers, half from SolidGen and half from DeepCAD. Each pair was independently judged by 7 crowd workers and we record the number of raters who identified the generated model as more realistic than randomly selected model from the training set. This gives us a "realism" score from 0–7 for each image pair. The percentage of image pairs with each realism are shown in the histogram in Figure 6 of the main paper.

If the human raters were selecting randomly between the two images then we would expect Figure 6 to form a binomial distribution. We see that for SolidGen the distribution is centered on a realism score of 3.5, but has wider tails than would be expected from chance alone. This indicates that while the realism of the

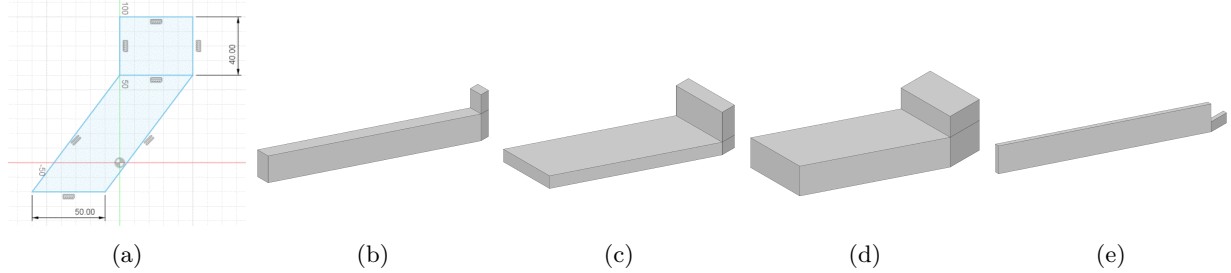

Figure A3: (a) An example of one of the parametric sketch templates used to build the dataset. (b–d) Examples of generated models with the extrusion from the inner wire added to the base extrusion. Here we see the effect of modifying the parameters controlling both the sketch geometry and the lengths of the two extrusions. (e) The result of a Boolean subtraction of the inner wire.

models generated by SolidGen is very similar to the training data, the raters do tend to agree on the realism of individual examples to a greater extent than would be expected by chance alone.

For the DeepCAD data there is a clear skew towards the raters identifying the training set as more realistic.

### A.5    Generation of the Parametric Variation Dataset

The parametric variation (PVar) dataset was created using constrained parametric sketches from the Sketch-Graphs dataset (Seff et al., 2020). Geometry and constraints were first extracted from 60 hand picked sketches and additional constraints and parameters controlling the lengths of lines and radii of arcs and circles added one by one until the sketches were optimally constrained (Bouma et al., 1995). These parameters could then be varied to generate multiple sketch geometries, while the constraints enforce aspects of the design intent such as horizontal, vertical and parallel lines and concentric arcs and circles.

In cases where the sketch geometry defined multiple adjacent regions (see Figure A3 (a)), the curves were organized into a set of nested wires. The outermost wire is chosen to contain the union of the regions and inner wires chosen at random to avoid any two wires sharing the same curve. The outermost wire is first extruded by a random distance forming a base extrusion. The inner wires are then extruded, starting from the top plane of the base extrusion and a Boolean union or subtraction is used to either add or remove material from the result. Figure A3 show the results of this process. The solids generated from each distinct sketch template are considered to belong to the same class resulting in 60 classes. Figure A4 shows one representative example from each class.

### A.6    Criteria used to Evaluate Validity of Generated B-reps

We consider a B-rep to be valid if it was successfully built from the network's output and additionally satisfies the following criteria:

1. **Triangulatable.**   Every face in the B-rep must generate at least one triangle, otherwise it cannot be rendered properly and is not possible to manufacture.

2. **Wire ordering.**   The edges in each wire in the B-rep must be ordered correctly. We check this using the `ShapeAnalysis_Wire::CheckOrder()` function in PythonOCC (Paviot, 2008) with a tolerance of 0.01.

3. **No wire self-intersection.**   The wires should not self-intersect to ensure that faces are well-defined and surfaces trimmed correctly. We check this using the `ShapeAnalysis_Wire::CheckSelfIntersection()` function in PythonOCC with a tolerance of 0.01.

4. **No bad edges.**   The edges in shells should be present once or twice but with different orientations. This can be checked with the `ShapeAnalysis_Shell::HasBadEdges()` function.

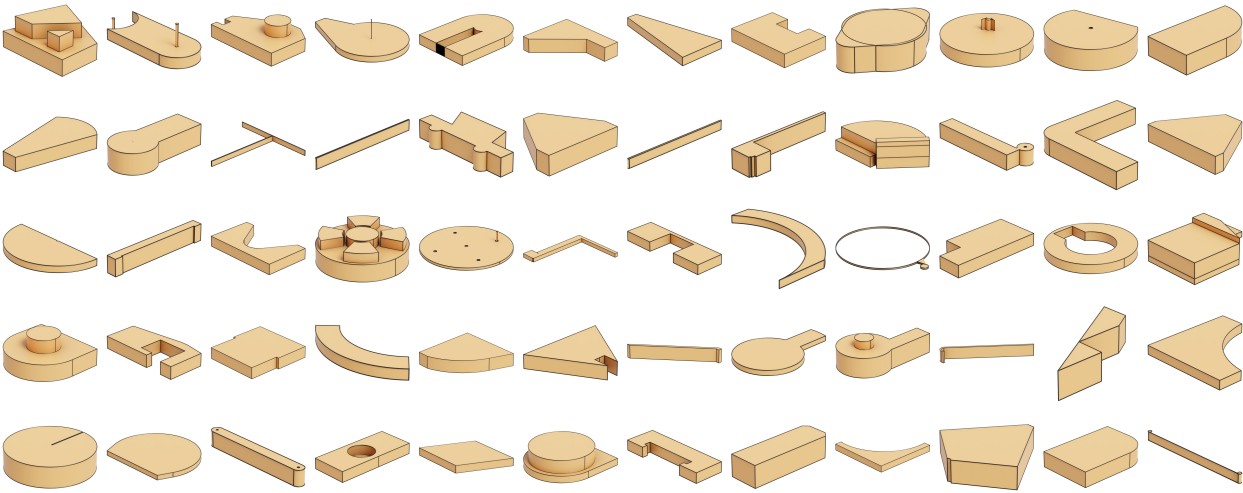

Figure A4: One example from each class in the PVar dataset.

## A.7 Architecture of Image and Voxel Encoders

The image encoder is a 2D convolutional neural network (CNN), that takes in $105 \times 128$ RGB images and outputs $(16 \times 16) \times d_{\text{emb}}$ features, where $d_{\text{emb}} = 256$. Its architecture is defined as:

$$\text{Conv2d}(64, 7, 3, 2) \rightarrow \text{GELU}() \rightarrow \text{Conv2d}(64, 3, 1, 2) \rightarrow \text{GELU}() \rightarrow \text{Dropout}(0.1) \rightarrow$$
$$\text{Conv2d}(64, 3, 1, 2) \rightarrow \text{GELU}() \rightarrow \text{Conv2d}(64, 3, 1, 1) \rightarrow \text{GELU}() \rightarrow \text{Dropout}(0.1) \rightarrow$$
$$\text{AdaptiveAvgPool2d}(16, 16) \rightarrow \text{Conv2d}(256, 3, 1, 1) \rightarrow \text{PosEmbed}() \rightarrow \text{SpatialFlatten}(),$$

where Conv2d(output channels, kernel size, padding, stride) is a 2D convolutional layer with bias included, GELU() is the Gaussian error linear unit activation (Hendrycks & Gimpel, 2016), Dropout(probability) is the dropout layer Srivastava et al. (2014), AdaptiveAvgPool2d(height, width) is an adaptive average pooling layer that outputs feature maps of the given spatial resolution, PosEmbed() learns embeddings for the spatial indices of the grid and adds them to the features, and SpatialFlatten flattens a convolutional feature map of shape $(N \times C \times H \times W)$ into $(N \times C \times (H \times W))$, and further reshapes it into $N \times (H \times W) \times C$ where the second dimension is treated as the sequence dimension as in Nash et al. (2020).

The voxel encoder is a 3D CNN that takes in $64 \times 64 \times 64$ binary voxel grids and outputs $(8 \times 8 \times 8) \times d_{\text{emb}}$ features. It's architecture is:

$$\text{Embed}(2, 8) \rightarrow \text{Conv3d}(64, 7, 3, 2) \rightarrow \text{GELU}() \rightarrow \text{Conv3d}(64, 3, 1, 1) \rightarrow \text{GELU}() \rightarrow \text{Dropout}(0.1) \rightarrow$$
$$\text{Conv3d}(64, 3, 1, 2) \rightarrow \text{GELU}() \rightarrow \text{Conv3d}(256, 3, 1, 1) \rightarrow \text{GELU}() \rightarrow \text{Dropout}(0.1) \rightarrow$$
$$\text{Conv3d}(256, 3, 1, 2) \rightarrow \text{PosEmbed}() \rightarrow \text{SpatialFlatten}(),$$

where the Embed(input dimension, output dimension) is a linear embedding layer that maps the binary voxels into learned 8 dimensional features. The output features are utilized by SolidGen via cross-attention as described in Section 5.2.

## A.8 Additional Results

Additional results comparing DeepCAD (Wu et al., 2021) and SolidGen on the unconditional generation task are shown in Figure A5 and Figure A6. We show additional results for class-conditional generation on the PVar dataset in Figure A7, and image conditional and voxel conditional results on the DeepCAD dataset in Figure A8 and Figure A9, respectively.

We include the top-$k$ accuracy plots ($1 \leq k \leq 10$) for SolidGen evaluated on the DeepCAD (Wu et al., 2021) test set in Figure A10.

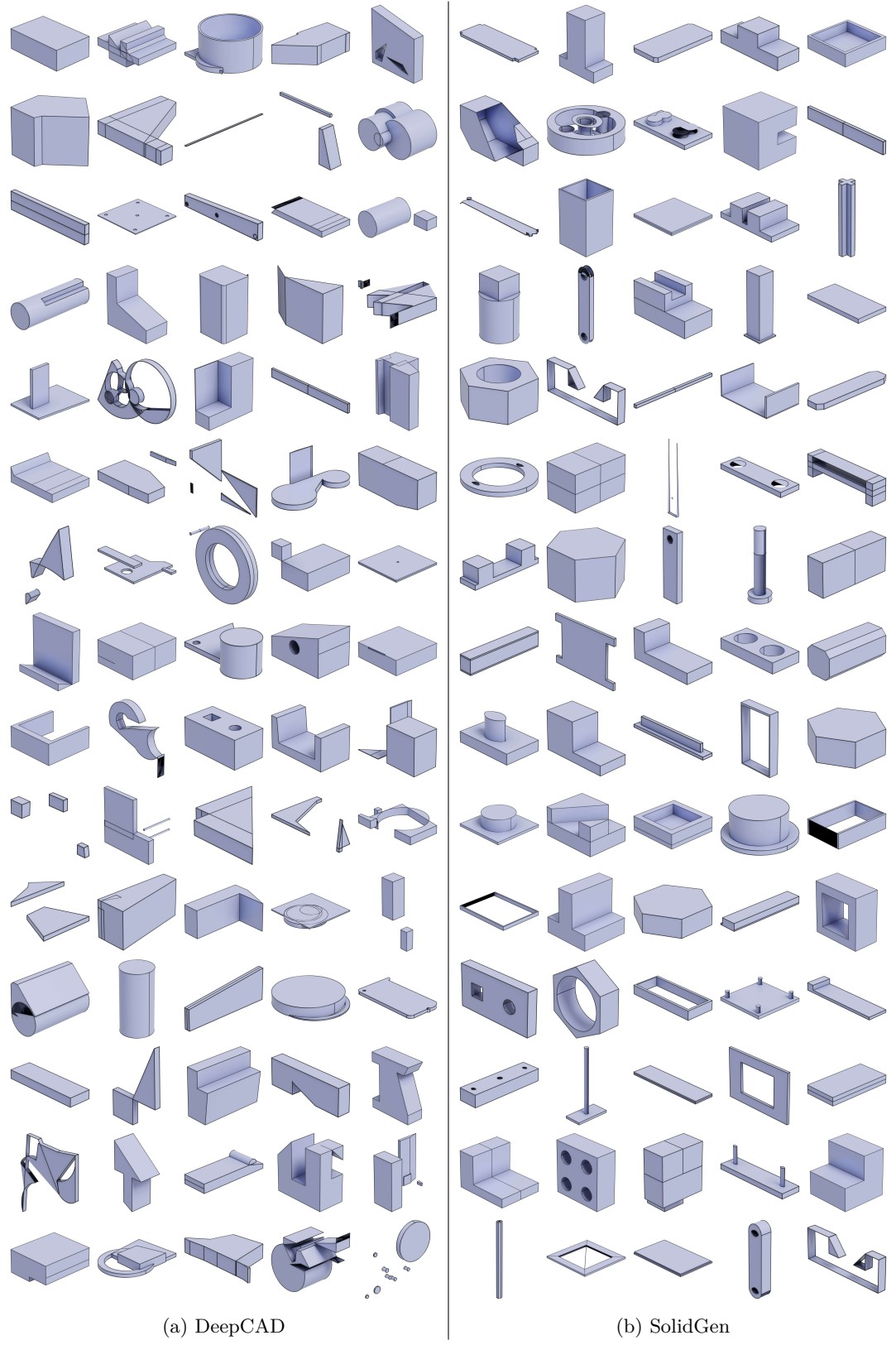

(a) DeepCAD          (b) SolidGen

Figure A5: Additional unconditional generation results.

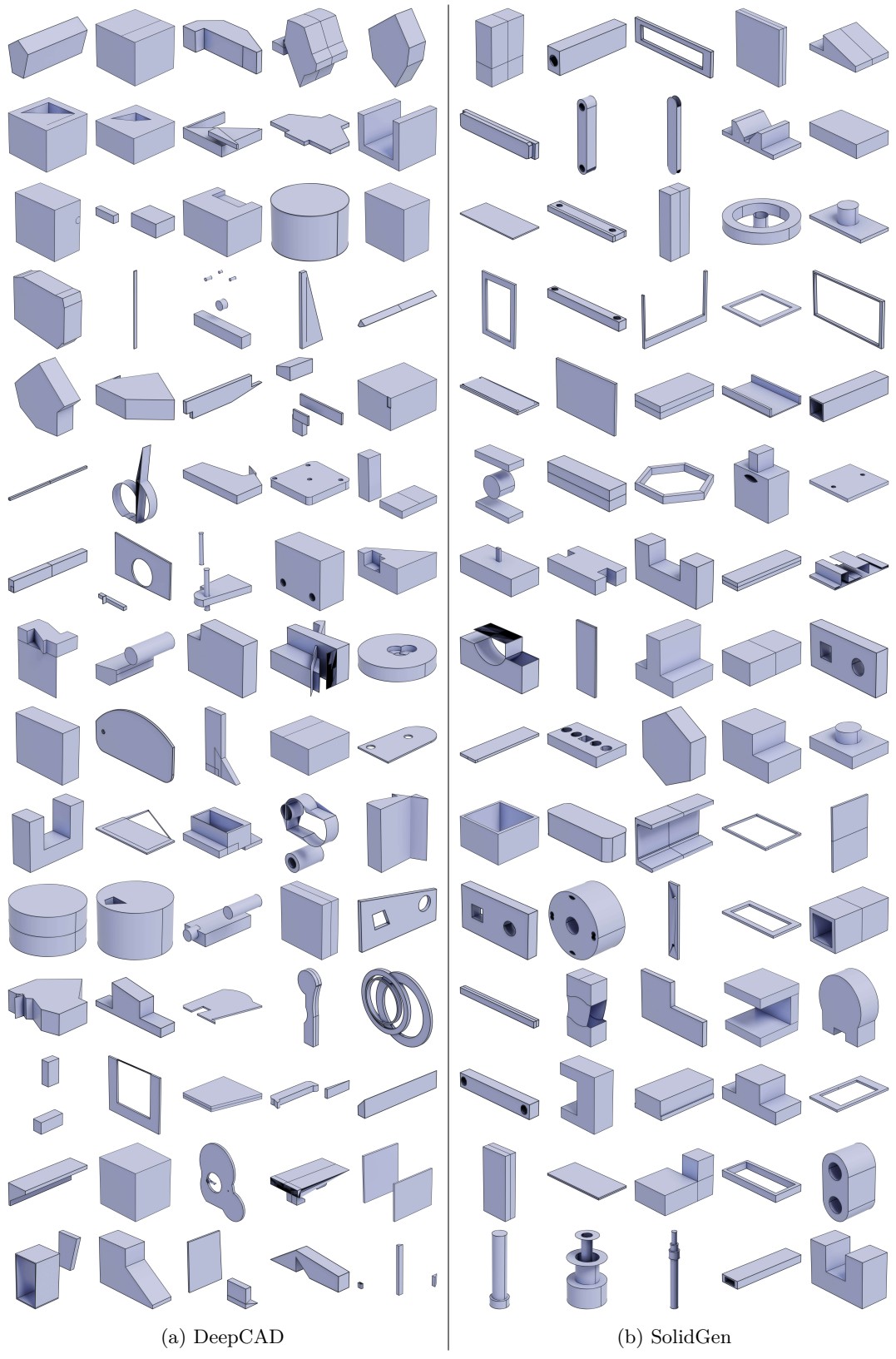

(a) DeepCAD                                             (b) SolidGen

Figure A6: Additional unconditional generation results.

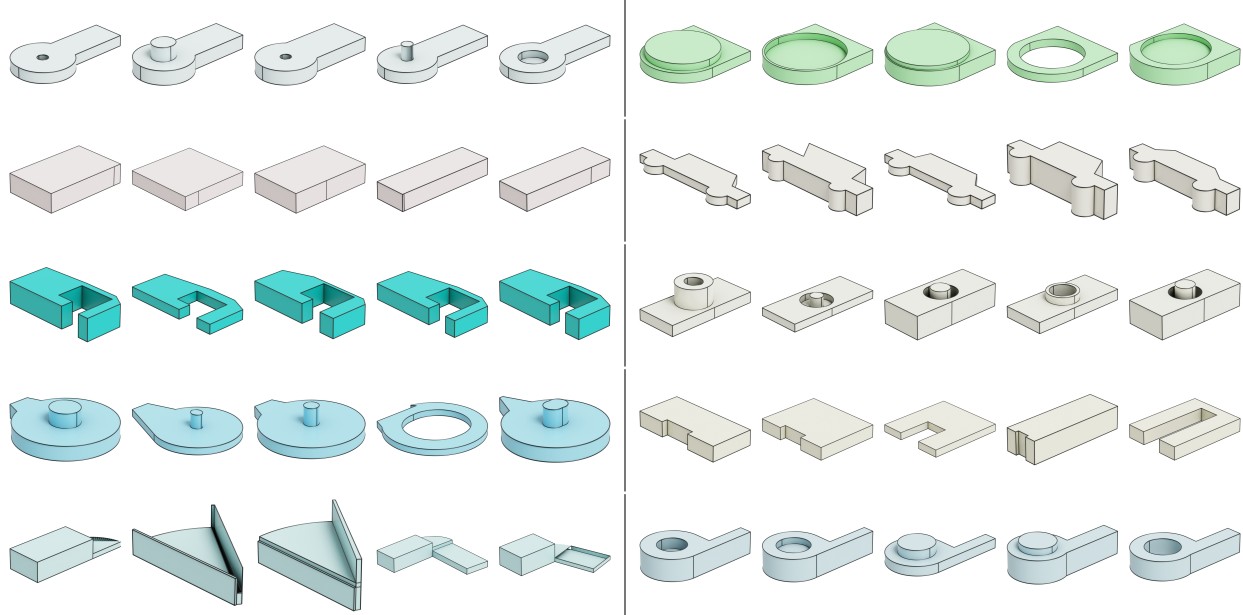

Figure A7: Additional class-conditional generation results. Each row shows samples generated from the same class.

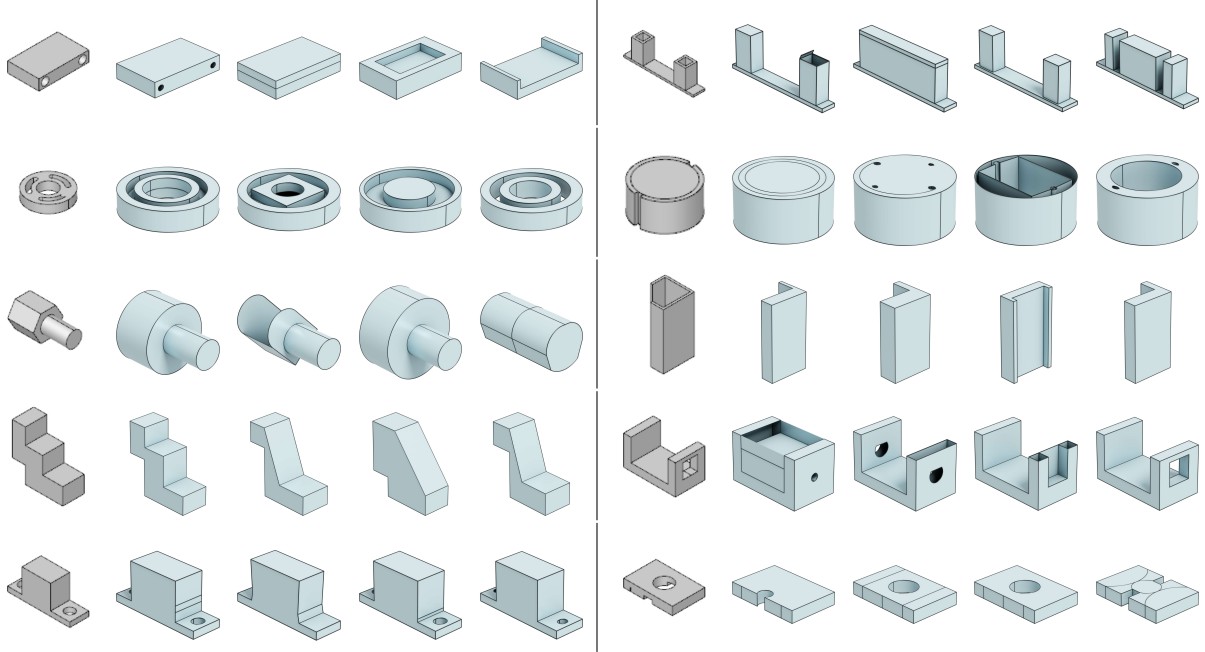

Figure A8: Additional image conditional samples from SolidGen using images (first column) obtained by rendering CAD models in the DeepCAD test set.

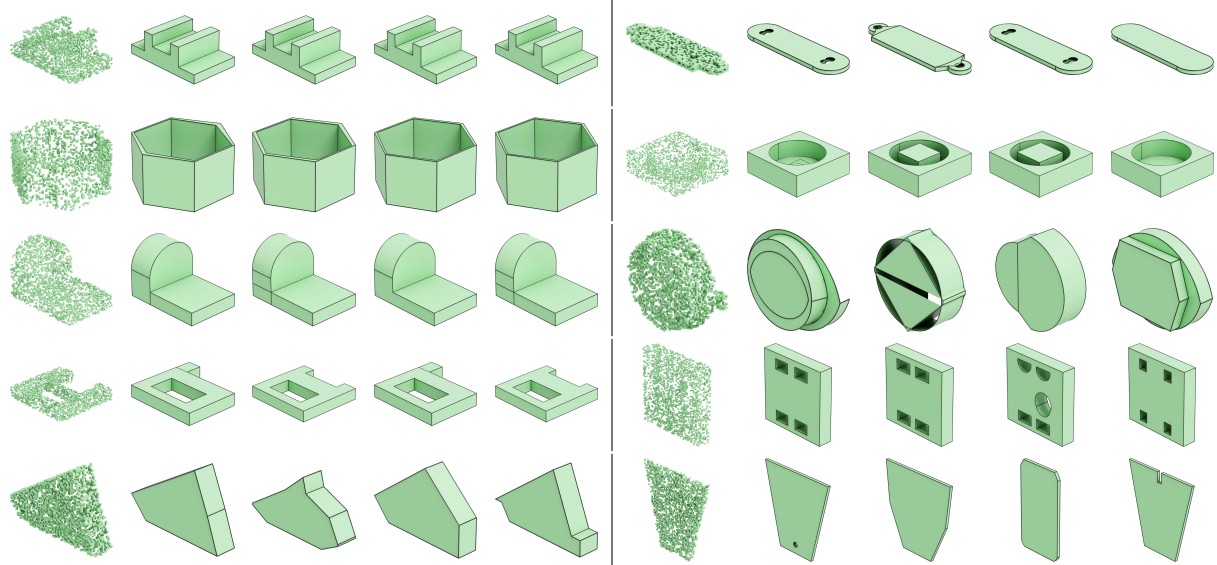

Figure A9: Additional voxel conditional samples from SolidGen using voxelized point clouds (first column) obtained by sampling 2048 points from CAD models in the DeepCAD test set.

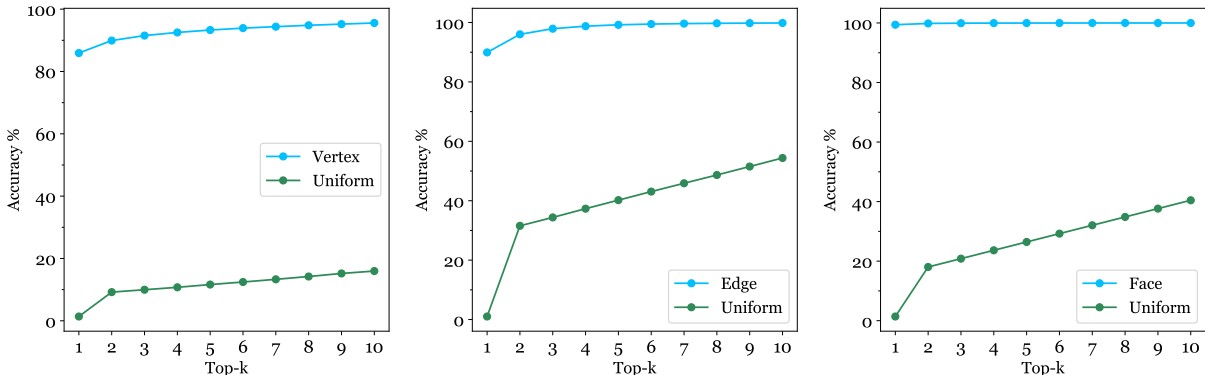

Figure A10: Top-$k$ next-token prediction accuracy of the unconditional SolidGen model trained on the DeepCAD dataset compared to a uniform baseline.

