# OpenReview forum: "SolidGen: An Autoregressive Model for Direct B-rep Synthesis"
_TMLR — Accepted by TMLR_

### Review · Reviewer_pQm5 · 2022-11-15

**Summary Of Contributions:**

The authors consider the problem of generating a CAD model through its B-Rep (boundary representation). The B-rep is the de-facto representation for parametric CAD models, representing shapes as geometric graphs. They propose an autoregressive model to generate the B-rep through its (hyper-)graph representation. The model uses a transformer architecture, augmented with pointer networks to represent relations. The model is evaluated on an unconditional generation task, as well as a generation task conditioned on a rendering of the CAD target.

**Audience:**

Yes

**Broader Impact Concerns:**

I do not have any broader impact concerns

**Claims And Evidence:**

Yes

**Requested Changes:**

1. It would be great to improve the description of the B-rep in section 3, by e.g. including diagrams as in fig. A1 as well as providing a formal mathematical description (perhaps simplified) of the data at hand. This would help make the main text better self-contained and more accessible to an applied ML audience who may not be familiar with CAD design. This would also clarify the constructions used in section 4. (This recommendation is important for my recommendation).

2. In fig. 3, the wireframe in the first panel is very light and hard to see - I would recommend increasing the line weight of the figure in order to make it more apparent. (This recommendation is minor)

3. In section 5, or a separate appendix, it would be great to include the following details of training: 1) all hyper-parameters of training, including AdamW L2 decay parameter, number of training epochs, learning rate scheduling (if any) etc. 2) include some details on how the hyper-parameters were selected (e.g. manually / automatically), 3) include some information about training time / hardware. (This recommendation is minor).

**Strengths And Weaknesses:**

## Strong points

The paper lays out a model which achieves its stated goals, and performs well on the evaluated problem. The proposed model is sound, and models the important relational aspect of the B-rep which is crucial for parametric CAD design. Overall, this paper provides an interesting demonstration of the possibility of auto-regressive generative modeling of B-rep for CAD design, with potentially interesting applications in the future.

---

> ### Author Response · Authors · 2023-01-10
> **Authors' Response to Review**
>
> Thank you for taking the time to review our paper and providing valuable feedback. Below are our responses to your requested changes.
> 1. > It would be great to improve the description of the B-rep in section 3, by e.g. including diagrams as in fig. A1 as well as providing a formal mathematical description (perhaps simplified) of the data at hand . This would help make the main text better self-contained and more accessible to an applied ML audience who may not be familiar with CAD design. This would also clarify the constructions used in section 4. (This recommendation is important for my recommendation).
>
> We have rewritten the “Boundary Representation” subsection in Section 3 and added more details and a new figure (Figure 2 in the revised paper) to comprehensively explain the B-rep data structure.
>
> 2. > In fig. 3, the wireframe in the first panel is very light and hard to see - I would recommend increasing the line weight of the figure in order to make it more apparent. (This recommendation is minor)
>
> We have increased the line weight.
>
> 3. > In section 5, or a separate appendix, it would be great to include the following details of training: 1) all hyper-parameters of training, including AdamW L2 decay parameter, number of training epochs, learning rate scheduling (if any) etc. 2) include some details on how the hyper-parameters were selected (e.g. manually / automatically), 3) include some information about training time / hardware. (This recommendation is minor).
>
> All hyperparameters, and hardware and training time information have been added to Section 5 under the "Implementation" subsection. They were all set to default values without any tuning, except the batch size which was chosen to maximize GPU usage.

---

### Review · Reviewer_7chE · 2022-11-21

**Summary Of Contributions:**


This paper proposes a new generative model, namely the SolidGen, that directly synthesizes the Boundary representation format (B-reps) from a sequence of CAD modeling operations in an unsupervised way using Transformers and two-level pointer networks. The SolidGen operates on the indexed boundary representation which is a new representation for B-reps as numeric arrays proposed by the authors. The index boundary representation maintains the geometry and topology of B-reps. The authors empirically show that the SolidGen achieve impressive results in both unconditional and conditional/controllable generation of B-reps.


**Audience:**

Yes

**Broader Impact Concerns:**

I have no concerns on the ethical implications of the work.

**Claims And Evidence:**

Yes

**Requested Changes:**

**Questions for the Authors:**

1. Can the authors provide comparisons to other methods on the task of generating B-reps?

2. How can the proposed method be combined with existing CAD methods or incorporated into existing CAD software such as AutoCAD? The authors should discuss these in the paper.

**Strengths And Weaknesses:**

###################################################################

**Summary of the Review:**

Overall, this paper could be an interesting contribution. There is no significant algorithmic development. However, the application studied in the paper is relevant, and the use of a transformer-based generative model to generate B-reps in this new application area is novel.

Currently, I am leaning toward accepting the paper.

###################################################################

**Strong points:**

1. The paper addresses an important problem in computer-aided design.

2. Utilizing a transformer-based generative model to generative B-reps is novel.

3. The empirical results in the paper are convincing.

4. The paper is well written with great-looking illustrative figures.

###################################################################

**Weak points:**

1. There is no significant algorithmic development in the paper.

2. Comparisons to other methods need to be provided.

---

> ### Author Response · Authors · 2023-01-10
> **Authors' Response to Review**
>
> Thank you for taking the time to review our paper and providing valuable feedback. Below are our responses to your requested changes.
>
> 1.  > Can the authors provide comparisons to other methods on the task of generating B-reps?
>
> To the best of our knowledge, SolidGen is the first generative model that can directly synthesize B-reps without the need for sketch-and-extrude sequence supervision. We have edited the contributions list in the introduction to clarify this. Since there were no other closely related published prior work, we compared with the state-of-the-art CAD generative model DeepCAD (ICCV 2021) that generates B-reps indirectly by generating the sketch-and-extrude sequence. The results are in Section 5.1 where we showed through quantitative and qualitative comparisons, as well as a user study, that SolidGen performs better while not requiring supervision from sketch-and-extrude sequences.
>
> 2.  > How can the proposed method be combined with existing CAD methods or incorporated into existing CAD software such as AutoCAD? The authors should discuss these in the paper.
>
> Thanks for this suggestion. All CAD software allows the import of solids without modeling history, so the B-reps produced by our method can be imported into them for further editing. We have added more discussion in the Conclusion section covering applications such as solid inpainting, sketch-based modeling, and data conversion.

---

### Review · Reviewer_84xv · 2022-12-17

**Summary Of Contributions:**

The paper proposes one of the first generative models for synthesizing solid 3D models structured as a boundary representation (B-rep). B-reps represent an important class of 3D representations that may be natively supported by a variety of CAD systems and geometric modelling kernels. Hence, the capability to directly synthesize data in B-rep format represents natural interest in the context of data-driven 3D design (e.g., for presenting designs initialised by scans, sketches or other forms of 2D/3D data).

The B-rep format considered in this project basically consists of three lists: vertices, edges (that index into vertices), and faces (index into edges). For each part of the format, a separate transformer-based model is trained to predict parameters of its respective instances in the list, given the previously generated sequence. I view the transformer model as an appropriate architecture due to its success in related tasks (e.g., mesh synthesis) and overall wide adoption.

In terms of quantitative evaluation, unconditional (from noise) and conditional (from shape category, image, and point cloud) generation results are presented; performance is described in terms of log-likelihood, B-rep validity, novelty, and uniqueness. While the unconditional samples are shown to be quite unique and diverse, constraining generation by feeding in an initializer image or point cloud correctly conditions the results to resemble the input. Both these qualities are important as creative tools aimed to help designers develop CAD designs.

**Audience:**

Yes

**Broader Impact Concerns:**

Most generative models require careful considerations of their generated samples to avoid ethical concerns. Another concern is leveraging large amounts of authored data in the context of product design (e.g., would one want to ensure that the design is fully “novel” so that no copyright issues are raised?). A recent Github Copilot issue illustrates the potential risks involved with using publicly gathered datasets in the context of data-driven design assistants.

**Claims And Evidence:**

Yes

**Requested Changes:**

I believe that the evaluation is sufficiently comprehensive, and the text is sufficiently well-written so that no critical changes to the submission is necessary.

**Strengths And Weaknesses:**

Strengths:

— The task that is being considered is new, relevant, and differs from those considered previously in the context of CAD generation (e.g. generating CSG trees or CAD commands).

— The solution is adequate and seems to be comprehensively addressing the problem.

— Qualitative and quantitative evaluations are presented versus one related work (DeepCAD) and convincing, with SolidGen models being generally more valid but arguably less diverse/novel.

— The text of the paper is clearly written and easy to follow.

— Interesting conditioning cases (e.g. images and point clouds) are presented.

Weaknesses:

— The list of unsupported CAD primitives includes certain important instances, e.g. B-splines which are arguably crucial in modern CAD designs.

— Technically, using pointer networks in the context of the considered task is similar to their applications in other works, e.g. PolyGen. However, this does not diminish the interesting application of these models to B-rep synthesis.

---

> ### Author Response · Authors · 2023-01-10
> **Authors' Response to Review**
>
> Thank you for your time and feedback.

---

### Review · Reviewer_8NAT · 2023-01-03

**Summary Of Contributions:**

The authors present the first autoregressive model of CAD designs that directly generates boundary-representation format (B-rep) designs. The work extends the PolyGen-style [1] modelling approach (autoregressive transformers + pointer networks) to B-rep data, formulating a data representation, and architecture modifications to handle this domain. The authors demonstrate that the approach works well and that generations can be conditioned by class, image and voxel contexts.

[1] PolyGen: An Autoregressive Generative Model of 3D Meshes, Charlie Nash, Yaroslav Ganin, S. M. Ali Eslami, Peter W. Battaglia, https://arxiv.org/abs/2002.10880

**Audience:**

Yes

**Broader Impact Concerns:**

I'm not aware of any broader impact concerns

**Claims And Evidence:**

Yes

**Requested Changes:**

Figure 1: Please be more explicit about whether the examples are data or samples. E.g. "two data examples", or "two samples"

Section 1: Please say how much bigger the datasets with / without modeling history are. This is an important point, and would save readers time.

End of section 2: "By contrast, our method is an autoregressive
generative model, and does not require an expensive optimization step to build plausible B-reps." -> autoregressive generation is pretty expensive too. Is there a more convincing argument for the benefits of autoregression vs the optimization-based approach?

Equation (3): The p(z) should be omitted, as we're conditioning on z

I would appreciate a bit more of an explanation of the use of B-rep arc points. Sec 4.1: "...additional points inserted on B-rep edges to encode the curve primitive information as explained in Figure 3", Figure 3 doesn't really explain how this works. How does it work with the position / coordinate embeddings?

Fig 7. Minor plotting issue: the bottom of the figures are cut off

Fig 9: It would be better to show the actual voxels, rather than point clouds




**Strengths And Weaknesses:**

Strengths:

* Direct B-rep modelling makes sense: Approaches that generate sequences of user actions can be constrained to a limited set of user operations. And it is complex to extend such approaches to the full range of useful operations. Also, data availability is an important consideration: Autoregressive modelling performance scales reliably with data, and CAD data withouts user operation sequences is more abundant.
* The work presents a sensible strategy for representating B-rep data in a way suitable for AR modelling. It builds on PolyGen and related models, with reasonable tweaks to the data representation and architecture.
* The paper is well-written: Visualizations are clear and useful, and the method description is clear.
* The results look good, although some instances of poorly formed designs can be found.

Weaknesses:

* The models are pretty small, which will impact performance. It would be interesting to see something like a scaling study: how performance changes as a function of model size. If overfitting is an issue at this dataset size, it would be good to know.

Questions:

* What is the impact of the 6-bit quantization? That isn't a whole lot of precision, does it limit the complexity of the shapes that can be generated?
* In the face model, does summing the vertex embeddings for an edge destroy order information which might be useful?
* Are the vertex encoders shared in the edge and face model? And if not, it could be worth trying.

---

> ### Author Response · Authors · 2023-01-10
> **Authors' Response to Review**
>
> Thank you for taking the time to review our paper and providing helpful suggestions. Below are our responses to your questions and requested changes.
>
> **Responses to questions:**
>
> 1. > What is the impact of the 6-bit quantization? That isn't a whole lot of precision, does it limit the complexity of the shapes that can be generated?
>
> We experimented with 6, 7 and 8-bit quantization and we found that 6-bit quantization was sufficient and led to faster convergence in the datasets we have considered. We believe this is due to the aligned nature of CAD data. We followed the lead of Seff et al. (Vitruvion: A Generative Model of Parametric CAD Sketches, ICLR 2022) who studied the effect of quantization on 2D CAD sketches and used 6-bit quantization. Xu et al. (SkexGen: Autoregressive Generation of CAD Construction Sequences with Disentangled Codebooks, ICML 2022) also used 6-bits with the DeepCAD dataset without significant loss in precision.
>
> 2. > In the face model, does summing the vertex embeddings for an edge destroy order information which might be useful?
>
> In our representation, the ordering of edge indices within each face is canonicalized by sorting them in ascending order since we wanted the face model to not be sensitive to the ordering of the edges. So, the sum operation is not a problem in practice.
>
> 3. > Are the vertex encoders shared in the edge and face model? And if not, it could be worth trying.
>
> Thanks for the interesting suggestion, we will try this in future. In our current implementation, the vertex encoder is not shared among the edge and face models.
>
> **Responses to requested changes:**
>
> 1. > Figure 1: Please be more explicit about whether the examples are data or samples. E.g. "two data examples", or "two samples"
>
> We have now explicitly mentioned that these are samples.
>
> 2. > Section 1: Please say how much bigger the datasets with / without modeling history are. This is an important point, and would save readers time.
>
> We have added the dataset sizes.
>
> 3. > End of section 2: "By contrast, our method is an autoregressive generative model, and does not require an expensive optimization step to build plausible B-reps." -> autoregressive generation is pretty expensive too. Is there a more convincing argument for the benefits of autoregression vs the optimization-based approach?
>
> We have added a sentence explaining how different input modalities e.g., class, image can be used with our model while ComplexGen requires a 3D point cloud. While autoregressive generation is indeed expensive, it is significantly faster than ComplexGen as it does not require an optimization routine.   SolidGen takes around 9 seconds on average per sample in the test set (including the postprocessing to convert the network’s outputs into B-reps), while ComplexGen takes an average of 9 minutes. Moreover, our representation is general and can be used without any changes in mask-based Transformers such as MaskGIT (CVPR 2022), where sampling is extremely fast due to parallel decoding.
>
> 4. > Equation (3): The p(z) should be omitted, as we're conditioning on z
>
> We have removed p(z).
>
> 5. > I would appreciate a bit more of an explanation of the use of B-rep arc points. Sec 4.1: "...additional points inserted on B-rep edges to encode the curve primitive information as explained in Figure 3", Figure 3 doesn't really explain how this works. How does it work with the position / coordinate embeddings?
>
> The arc points are not treated differently from the rest of the B-rep vertices. We apply the same positional and coordinate embeddings for these points like the other vertices. The arc midpoint is treated specially only during the conversion of the network’s output (indexed B-rep) to an actual B-rep. In this process, the arc midpoint is used to determine the geometry of the arc only and does not get converted into a B-rep vertex. We discuss this in Appendix A.2.1.
>
> 6. > Fig 7. Minor plotting issue: the bottom of the figures are cut off
>
> We have fixed the cropping.
>
> 7. > Fig 9: It would be better to show the actual voxels, rather than point clouds
>
> Here we are indeed showing the voxels by visualizing each active voxel as a cube. The voxel model represents the occupancy of the point cloud. Each voxel is set to 1 if a quantized point from the point cloud lies within it, and 0 otherwise.

---

### Decision · Action_Editors · 2023-01-26

**Recommendation:** Accept with minor revision

**Comment:**

The paper is of particularly high-quality: well-written, technically sound, making a novel and potentially impactful contribution that is convincingly evaluated. I think it's appropriate for a featured certification.

Before the paper can be formally accepted, I would like to request a minor revision. In the camera-ready version, I would like the authors to comment on the following questions, which came up during the discussion with the reviewers:

1. The list of unsupported CAD primitives includes certain instances, such as B-splines, which are important in modern CAD. Can the proposed approach be modified to take these into account?

2. The paper considers relatively small models in the experimental evaluation. How does the approach scale with model size? Is overfitting a concern?


**Audience:**

In the opinion of the reviewers, the paper proposes an effective approach for a potentially impactful task. The paper pushes the state-of-the-art in generation of computed-aided designs. I believe the paper would be interesting not only to a CAD-modelling audience, but also to the generative-modelling community more broadly.

**Claims And Evidence:**

The paper proposes a generative model of computer-aided designs (CAD) that operates directly on the boundary-representation format (B-rep). The model is autoregressive and based on transformers and pointer networks, in the style of PolyGen. The generation capabilities of the model are thoroughly and rigorously evaluated, using a number of objective metrics and human perceptual evaluation. All four reviewers were convinced by the empirical evaluation.

---

> ### Author Response · Authors · 2023-02-03
> **Authors' Response to Action Editors**
>
> We thank the editors and the reviewers for the overwhelmingly positive feedback, valuable suggestions, and recognizing our paper with the featured certification. Please find our detailed responses to the questions below. We have incorporated these responses into the camera-ready paper in the “Implementation” paragraph under Section 5 (lines 9—11), and the “Limitations & Future Work” paragraph in Section 6 (lines 6—9).
>
> > 1. The list of unsupported CAD primitives includes certain instances, such as B-splines, which are important in modern CAD. Can the proposed approach be modified to take these into account?
>
> Supporting B-spline curves and surfaces is something we are keen to try in future. It is possible to support fixed-degree uniform B-spline curves in our framework by defining any edge in the indexed B-rep representation that groups >= (degree+1) vertices to be considered as representing a spline. Supporting spline surfaces would require predicting a grid of interpolating points or control points, as the B-spline surface parameters cannot be fully determined by the boundary curves, unlike prismatic surfaces.
>
> > 2. The paper considers relatively small models in the experimental evaluation. How does the approach scale with model size? Is overfitting a concern?
>
> Overfitting was certainly a concern given the relatively small size of our datasets. Previous sketch-and-extrude style generative models (DeepCAD, SkexGen) trained on the DeepCAD dataset have used 4-layer Transformer models. In initial experiments, we tried using 4-layer, 8-layer and 12-layer models and found 8-layer models to perform reasonably without too much under/over-fitting. Scaling our model and training it on larger datasets like ABC is a natural next step we are keen to explore in the near future.